# PERFORMANCE-ENHANCED AGGREGATED REPRESENTATION LEARNING

## ABSTRACT

Representation learning is a key technique in modern machine learning that enables models to identify meaningful patterns in complex data. However, different methods tend to extract distinct aspects of the data, and relying on a single approach may overlook important insights relevant to downstream tasks. This paper proposes a performance-enhanced aggregated representation learning method, which combines multiple representation learning approaches to improve the performance of downstream tasks. The framework is designed to be general and flexible, accommodating a wide range of loss functions commonly used in machine learning models. To ensure computational efficiency, we use surrogate loss functions to facilitate practical weight estimation. Theoretically, we prove that our method asymptotically achieves optimal performance in downstream tasks, meaning that the risk of our predictor is asymptotically equivalent to the theoretical minimum. Additionally, we derive that our method asymptotically assigns nonzero weights to correctly specified models. We evaluate our method on diverse tasks by comparing it with advanced machine learning models. The experimental results demonstrate that our method consistently outperforms baseline methods, showing its effectiveness and broad applicability in real-world machine learning scenarios.

## 1 INTRODUCTION

The performance of machine learning methods heavily relies on the choice of data representations (Bengio et al., 2013). From traditional principal component analysis (PCA) to modern deep learning techniques, representation learning methods have become increasingly deep and complex in pursuit of better task performance (Xie et al., 2020). As a result, a large body of work has proposed diverse approaches to learning effective representations. Some studies have focused on selecting the most suitable representations for a specific downstream task (Nguyen et al., 2020; You et al., 2021), thereby leading to a model selection problem. However, relying on a single representation learning method is risky since no single representation learning method is uniformly optimal across tasks, data regimes, or modalities. First, different representation learners specialize in different types of information. For instance, in pretrained language models, different architectures tend to encode complementary syntactic and semantic signals across models and layers (Liu et al., 2019). Second, empirical studies demonstrate that representation quality varies substantially across tasks and distributions, and representations that perform well in one setting may fail in another (Ericsson et al., 2021; Agostinelli et al., 2022). To address this issue, the idea of combining heterogeneous representations has proven success in multi-view, multi-modal, and ensemble learning literature (Lin et al., 2017; Li & Li, 2022).

However, most existing representation aggregation techniques either simply fuse all available representations (e.g., by concatenation) or treat each representation set as equally important, implicitly assuming that they are equally informative for the downstream task. In practice, however, different representation sets can contribute unequally to the task. Some methods attempt to address this by learning data-dependent weights over representations using complex black-box modules. However, these approaches tend to be hard to interpret and lack theoretical guarantees on why the learned combination improves over its constituents. To overcome these issues, we propose a model averaging framework called performance-enhanced aggregated representation learning (PEARL), designed to improve downstream performance by combining diverse representations. Model averaging is a

popular technique that addresses the uncertainty of choosing models (Hjort & Claeskens, 2003; Hansen, 2007), and recent works have demonstrated its success in complex models including random forests (Chen et al., 2024) and neural networks (Zeng & Van den Broeck, 2023; Wang et al., 2025). However, model-averaging methods with rigorous theoretical guarantees have largely been developed for classical statistical models (e.g., linear or quasi-likelihood models) under convex and smooth loss functions. These settings do not directly cover heterogeneous deep architectures and non-convex, task-specific objectives that arise in modern complex tasks. This gap motivates a more general model-averaging framework that can accommodate diverse deep models and complicated loss structures. PEARL addresses this by using representation learning as a bridge from traditional statistical models to advanced deep methods, providing theoretically grounded guidance for combining multiple representation learners.

PEARL consists of two stages. In the first stage, a set of foundation representation learning (FRL) models, which are either trained from scratch or adapted from pretrained encoders such as BERT (Devlin et al., 2019) and GPT (Brown et al., 2020), transform raw inputs into representation vectors. These representations, individually or combined through integration strategies, form a pool of candidate representation sets. In the second stage, each candidate set is linked to the target task through a downstream predictor (e.g., a linear model or neural network). The final output of PEARL is obtained by averaging these predictions, with weights estimated by a cross-validation-based weighting criterion.

The term "performance-enhanced" in PEARL reflects its effectiveness in both computational feasibility and theoretical guarantees. For the computational simplicity, a key design principle of PEARL is the weighting procedure. To accommodate diverse downstream tasks, the weighting criterion is formulated on the basis of a generalized loss function. However, directly optimizing certain loss functions may introduce non-smooth or non-convex optimization challenges. To enable efficient and stable weight computation, PEARL employs surrogate loss functions as approximations to original loss functions. Moreover, PEARL adopts a linear weighting scheme rather than nonlinear deep aggregation, thereby further reducing computational cost and preserving interpretability.

We establish two main theoretical guarantees for PEARL. First, from the perspective of downstream task performance, we prove the asymptotic optimality of the PEARL predictor. Specifically, the predictive risk of PEARL converges to the lowest possible risk defined by the original loss. This result ensures that PEARL performs at least as well as any single FRL in the pool and other linear combinations of FRLs, such as simple ensemble. Second, from the perspective of weight interpretability, we prove weight consistency. Specifically, when the candidate pool contains correctly specified models, PEARL asymptotically assigns all weight to them. This property demonstrates that PEARL is able to identify and prioritize well-specified models among candidates. Together, these results provide a solid theoretical foundation for both the effectiveness and the interpretability of PEARL across diverse downstream tasks.

## 2 METHOD

Given predictors predictors $x \in \mathbb{X}$ and outputs $y \in \mathbb{Y}$. We aim to predict an output in a downstream task for a new input. The predictor space $\mathbb{X}$ is general and may include structured elements (e.g., vectors) and unstructured elements (e.g., text, images).

### 2.1 REPRESENTATION LEARNING PROCESS

A representation learning function $f$ maps each predictor $x \in \mathbb{X}$ to a $p$-dimensional representation vector $\boldsymbol{z} \in \mathbb{R}^p$, i.e., $f : \mathbb{X} \to \mathbb{R}^p$, $x \mapsto \boldsymbol{z}$. Representation learning methods vary based on the type of input data and the analyst's preferences. For structured data, techniques such as PCA are commonly used for dimensionality reduction. For image data, methods like convolutional autoencoders (AE) or contrastive learning approaches (e.g., SimCLR (Chen et al., 2020)) have proven effective in extracting meaningful representations. For text data, models such as Word2Vec or transformer-based approaches (e.g., BERT) are frequently employed to generate semantic embeddings. Moreover, each method involves specific hyperparameters (e.g., the number of principal components in PCA, network architecture details in autoencoder or transformer models) that can lead to variations in the learned representations, thereby constituting different representation learning techniques.

We consider $M$ representation learning models as our FRL models, each mapping the input space $\mathbb{X}$ to a model-specific $p_m$-dimensional vector space $\mathbb{R}^{p_m}$, i.e., for the $m$-th model,

$$f_m : \mathbb{X} \to \mathbb{R}^{p_m}, \quad x \mapsto \boldsymbol{z}_{(m)}.$$

The function space is defined by $\mathbb{F}$. These models are trained on the unlabeled dataset $\mathcal{D}^{\mathrm{u}} = \{x_i' \in \mathbb{X}\}_{i=1}^N$, where $x_i'$ is the $i$-th unlabeled sample and $N$ denotes the number of unlabeled samples. Suppose the $m$-th FRL model employs a representation learning loss $U_m(\cdot)$ on $\mathcal{D}^{\mathrm{u}}$ to obtain

$$\widehat{f}_m = \arg\min_{f \in \mathbb{F}} \sum_{i=1}^N U_m(f(x_i')).$$

For example, let us illustrate the PCA process to extract representations for panel data. Define the data matrix by $\boldsymbol{X}' = (\boldsymbol{x}_1', \dots, \boldsymbol{x}_N')^\top \in \mathbb{R}^{N \times T}$, where $\boldsymbol{x}_i'$ is the feature vector of the $i$-th observation. PCA seeks a linear map that gets the $p_m$-dimensional latent factors: $\boldsymbol{z}_{i,(m)}' = f_m(\boldsymbol{x}_i') = \boldsymbol{B}_m^\top \boldsymbol{x}_i'$, where $\boldsymbol{B}_m \in \mathbb{R}^{T \times p_m}$ is the parameter matrix. Let $\boldsymbol{Z}_{(m)}' = (\boldsymbol{z}_{1,(m)}', \dots, \boldsymbol{z}_{n,(m)}')^\top$. Then $\widehat{f}_m(\boldsymbol{x}) = \widehat{\boldsymbol{B}}_{(m)}^\top \boldsymbol{x}$ for any vector $\boldsymbol{x} \in \mathbb{X}$, where $\widehat{\boldsymbol{B}}_{(m)}$ is calculated by solving a constrained least squares loss function after centering the columns of $\boldsymbol{X}'$:

$$(\widehat{\boldsymbol{Z}}_{(m)}', \widehat{\boldsymbol{B}}_m) = \arg\min_{\boldsymbol{Z}, \boldsymbol{B}} \left( \|\boldsymbol{X}' - \boldsymbol{Z}\boldsymbol{B}^\top\|_F^2 \right),$$

$$\text{s.t. } N^{-1} \sum_{i=1}^N \boldsymbol{z}_i \boldsymbol{z}_i^\top = \boldsymbol{I}_{p_m} \text{ and } \boldsymbol{B}^\top \boldsymbol{B} \text{ diagonal,}$$

where $\boldsymbol{z}_i$ is the $i$-th row of $\boldsymbol{Z}$ and $\widehat{\boldsymbol{Z}}_{(m)}'$ is the estimator of $\boldsymbol{Z}_{(m)}'$. The loss $U_m(\cdot)$ can be a $U$-embedding loss function designed for unannotated data (Dai et al., 2022) and can be tailored for different scenarios including contrastive loss or generative modeling objectives (Chen et al., 2020; He et al., 2020; Kingma & Welling, 2014). In each case, the structure of $f_m$ and the dimension $p_m$ in the resulting representations might differ.

## 2.2 DOWNSTREAM TASK ANALYSIS

In this subsection, we focus on the downstream task analysis on a labeled dataset $D^l = \big\{(x_i, y_i) \in \mathbb{X} \times \mathbb{Y}\big\}_{i=1}^n$, where $n$ is the number of labeled samples. Specifically, this subsection consists of two parts: constructing a set of candidate representations and developing a model averaging predictor to estimate the output for a new input $x_{\mathrm{new}} \in \mathbb{X}$.

With $M$ learned FRL models $\{\widehat{f}_1, \widehat{f}_2, \dots, \widehat{f}_M\}$, each $\widehat{f}_m$ maps $x_i \in \mathbb{X}$ to a vector in $\mathbb{R}^{p_m}$. Symbolically,

$$\widehat{\boldsymbol{z}}_{i,(m)} = \widehat{f}_m(x_i) \text{ for } i = 1, \dots, n, \ , m = 1, \dots, M,$$

where $\widehat{\boldsymbol{z}}_{i,(m)} \in \mathbb{R}^{p_m}$ is defined as the foundation representation vector under the $m$-th FRL model. By collecting these vectors for all $n$ samples, we form the foundation representation matrix $\widehat{\boldsymbol{Z}}_{(m)} = (\widehat{\boldsymbol{z}}_{1,(m)}, \dots, \widehat{\boldsymbol{z}}_{n,(m)})^\top \in \mathbb{R}^{n \times p_m}$ for the $m$-th FRL model.

Next, we introduce candidate representation sets that include both the foundation representations directly learned by FRL models and additional representations obtained through fusion or combinatorial techniques. To avoid notational confusion, with a total of $J$ candidate representation sets ($J \geq M$), for $1 \leq i \leq n$, we denote the $j$-th candidate representation vector by $\widehat{\boldsymbol{z}}_{i,[j]}$ and their corresponding matrices $\widehat{\boldsymbol{Z}}_{[j]} = (\widehat{\boldsymbol{z}}_{1,[j]}, \dots, \widehat{\boldsymbol{z}}_{n,[j]})^\top$ for $1 \leq j \leq J$. Note that the subscript bracket for notations of candidate representation sets is $[]$ and is different from $()$ for the foundation representation set $\widehat{\boldsymbol{z}}_{i,(m)}$ for $1 \leq m \leq M$. Let $p_j$ be the dimension of the vector $\widehat{\boldsymbol{z}}_{i,[j]}$ and therefore $\widehat{\boldsymbol{Z}}_{[j]} \in \mathbb{R}^{n \times p_j}$.

For additional representation sets, one simple way is to consider a simple fusion technique that concatenates the foundation representations: $(\widehat{\boldsymbol{z}}_{i,(1)}^\top, \dots, \widehat{\boldsymbol{z}}_{i,(M)}^\top)^\top \in \mathbb{R}^p$, where $p = \sum_{m=1}^M p_m$. The corresponding fusion matrix is $(\widehat{\boldsymbol{Z}}_{(1)}, \dots, \widehat{\boldsymbol{Z}}_{(M)})$. This approach is common in deep learning to integrate complementary information from various models (Zhang et al., 2020). In addition to simple fusion, we provide Example 1 below to illustrate several methods for constructing additional representation sets.

**Example 1.** *Some methods for generating candidate representation sets when there are $M$ FRL models $\{\widehat{f}_1, \widehat{f}_2, \ldots, \widehat{f}_M\}$:*

1. *Foundation Set Combinations. For instance, when $M = 3$ and each $p_m = 2$, using all non-empty subsets of the three foundation representation sets yields $J = 2^M - 1 = 7$ candidate representation sets. This includes individual foundation sets, pairwise concatenations, and the complete concatenation of all three. Mathematically, the candidate representation sets are $\widehat{\boldsymbol{Z}}_{[j]} = \widehat{\boldsymbol{Z}}_{(j)}$ for $j = 1, 2, 3$; $\widehat{\boldsymbol{Z}}_{[4]} = (\widehat{\boldsymbol{Z}}_{(1)}, \widehat{\boldsymbol{Z}}_{(2)})$; $\widehat{\boldsymbol{Z}}_{[5]} = (\widehat{\boldsymbol{Z}}_{(1)}, \widehat{\boldsymbol{Z}}_{(3)})$; $\widehat{\boldsymbol{Z}}_{[6]} = (\widehat{\boldsymbol{Z}}_{(2)}, \widehat{\boldsymbol{Z}}_{(3)})$; and $\widehat{\boldsymbol{Z}}_{[7]} = (\widehat{\boldsymbol{Z}}_{(1)}, \widehat{\boldsymbol{Z}}_{(2)}, \widehat{\boldsymbol{Z}}_{(3)})$.*

2. *Fusion Matrix Columns Combinations. For instance, when $M = 3$ and each $p_m = 2$, concatenating all foundation representations forms a fusion matrix with $p = 6$ columns. Taking every non-empty subset of these columns gives $J = 2^p - 1 = 63$ candidate representation sets. Each combinatorial subset of the six columns in the fusion matrix forms a candidate representation set. Although this method yields a larger set of candidates, it also poses challenges in terms of computational complexity and potential redundancy.*

3. *Domain knowledge-based selection. Domain knowledge can be encoded by designating some FRLs as "always-on" and others as "optional". For example, in asset pricing, well-established asset factors may be combined with factors learned via deep learning to enhance interpretability and performance (Feng et al., 2024). This approach helps mitigate the combinatorial explosion in candidate sets by narrowing the focus to the most relevant representations.*

After we construct the $J$ candidate representation sets, we build the relationship between candidate representations and the output on the labeled dataset $\mathcal{D}^l$. For the $j$-th candidate representation set, a function $g_j$ that maps $\mathbb{R}^{p_j}$ onto $\mathbb{Y}$ is:

$$g_j : \widehat{\boldsymbol{z}}_{i,[j]} \mapsto y_i \text{ and } g_j \in \mathbb{G},$$

where $\mathbb{G}$ is the class of considered functions. Depending on the problem type, the model $g_j$ can take various forms, such as linear models, decision trees (Breiman et al., 1984), and other machine learning models like neural networks. The function $g_j$ is learned using $(\widehat{\boldsymbol{z}}_{i,[j]}, y_i)_{i=1}^n$ and the corresponding estimated function is defined by $\widehat{g}_j$ for $1 \leq j \leq J$. When the dimensionality of the representations is too large, $g_j$ can be a sparsity-inducing function to reduce the dimensionality. For example, the regularization method in (Tibshirani, 1996; Fan & Li, 2001) can be utilized.

Consider a new observation $x_{\text{new}} \in \mathbb{X}$ and our goal is to predict its label $y_{\text{new}}$. We first generate $M$ foundation representation sets by applying $M$ FRL models on $x_{\text{new}}$, and then construct $J$ candidate representation sets for $x_{\text{new}}$, denoted as $\{\widehat{\boldsymbol{z}}_{\text{new},[j]}\}_{j=1}^J$. For each $1 \leq j \leq J$, the candidate model generates an output as follows:

$$\widehat{y}_{\text{new},[j]} = \widehat{g}_j(\widehat{\boldsymbol{z}}_{\text{new},[j]}).$$

We propose a model averaging approach to aggregate these $J$ outputs by assigning weights to each of them. Specifically, the model averaging predictor is defined as:

$$\widehat{y}_{\text{new}}(\boldsymbol{w}) = \sum_{j=1}^J w_j \widehat{y}_{\text{new},[j]}, \tag{1}$$

where $w_j$ is the weight assigned to the $j$-th output, and the weight vector $\boldsymbol{w} = (w_1, \ldots, w_J)^{\text{T}}$ lies in the simplex $\mathbb{W}$ satisfies that $\boldsymbol{w} \in \mathbb{W} = \{\boldsymbol{w} \in [0, 1]^J : \sum_{j=1}^J w_j = 1\}$. Next, we discuss how to tune the weights for different downstream tasks.

### 2.3 WEIGHT CHOICE CRITERION

For downstream tasks, the criteria for model evaluation and their corresponding loss functions often differ. Our paper aims to build a generalized framework that can adaptively handle different kinds of loss functions based on various machine-learning-based candidate models. A simple but naive approach is to construct the weight criterion that is directly based on the original loss function. The weights are tuned on the labeled dataset by

$$\widehat{\boldsymbol{w}} = \underset{\boldsymbol{w} \in \mathbb{W}}{\arg\min} \, n^{-1} \sum_{i=1}^n L(y_i, \widehat{y}_i(\boldsymbol{w})), \tag{2}$$

where $\widehat{y}_i(\boldsymbol{w}) = \sum_{j=1}^J w_j \widehat{y}_{i,[j]}$ and $L(\cdot, \cdot)$ is a loss function that measures the difference between the true responses and the weighted predictions. However, the weighting criterion equation 2 has two problems. First, many loss functions, particularly in complex or classification problems, can be non-convex or non-differentiable (e.g., the 0-1 loss or certain robust loss functions). These properties hinder the direct application of gradient-based optimization methods, potentially leading to suboptimal solutions or convergence issues. Second, optimizing the loss over the all training samples may cause the model to fit the noise or idiosyncratic patterns in the training data rather than capturing the underlying general patterns. This may increase the risk of overfitting problem (Hastie et al., 2009).

For the computational complexity problem in equation 2, surrogate loss functions that are convex and differentiable are often employed. For some surrogate losses, their connection to the original loss is supported by theoretical guarantees, such as classification calibration, consistency in risk minimization, or upper bounds on the excess risk measured by the original loss (Mao et al., 2023). For example, in classification task, the hinge loss (Cortes & Vapnik, 1995), the logistic loss (Zhu & Hastie, 2005), and the $\psi$-loss (Shen et al., 2003) are often employed as smooth approximations for the zero-one classification loss. These surrogate losses not only facilitate optimization but also have been shown to preserve desirable theoretical properties under certain conditions, such as minimizing the surrogate loss leads to minimization of the original loss under suitable conditions. In addition, if the original loss function is computational friendly, such as the mean squared error in regression tasks, it is entirely reasonable to use the original loss function without substitution.

For the overfitting problem in equation 2, we propose a cross-validation-based weight tuning algorithm (Algorithm 1). Let $\mathbb{S} = \{1, \ldots, n\}$ denote the index set, which is divided into $K$ equally sized folds: $\mathbb{S} = \bigcup_{k=1}^K \mathbb{S}_k$, where $\mathbb{S}_k$ is the index set in the $k$-th fold. For the $j$-th candidate model and each $k \in \{1, \ldots, K\}$, we refit the model $g_j$ using the dataset $\{(\widehat{\boldsymbol{z}}_{i,[j]}, y_i)\}_{i \in \mathbb{S} \setminus \mathbb{S}_k}$. The resulting refitted function of $g_j$ excluding the $k$-fold is denoted by $\tilde{g}_j^{(-k)}$. Then $\tilde{g}_j^{(-k)}$ is applied on $\{\widehat{\boldsymbol{z}}_{i,[j]}\}_{i \in \mathbb{S}_k}$ to obtain $\tilde{y}_{i,[j]} = \tilde{g}_j^{(-k)}(\boldsymbol{z}_{i,[j]})$ with $i \in \mathbb{S}_k$. Finally, we estimate the weights by

$$\widehat{\boldsymbol{w}} = \operatorname{argmin}_{\mathbf{w} \in \mathbb{W}} n^{-1} \sum_{i=1}^n V(y_i, \widetilde{y}_i(\boldsymbol{w})), \tag{3}$$

where $V$ is the surrogate loss function for the original loss function $L$, $\widehat{\boldsymbol{w}} = (\widehat{w}_1, \ldots, \widehat{w}_J)^\top$, and $\widetilde{y}_i(\boldsymbol{w}) = \sum_{j=1}^J w_j \widetilde{y}_{i,[j]}$.

By equation 3, we plug in the obtained $\widehat{\boldsymbol{w}}$ in equation 1 to derive our PEARL prediction for the label of the new observation $x_{\text{new}}$:

$$\widehat{y}_{\text{new}}(\widehat{\boldsymbol{w}}) = \sum_{j=1}^J \widehat{w}_j \widehat{y}_{\text{new},[j]}. \tag{4}$$

Furthermore, PEARL can be extended to multi-modal data, which is presented in the Appendix A.

## 3  THEORETICAL RESULTS

In this section, we demonstrate the theoretical efficiency of PEARL by asymptotic optimality and weight consistency.

For the representation learning process, let $f_m^*$ represent the optimal representation learning function that maps from $\mathbb{X}$ to $\mathbb{R}^{p_m}$ under the given representation learning loss $U_m(\cdot)$, that is $f_m^* = \arg\min_{f \in \mathbb{F}} \mathbb{E}[U_m(f(x))]$ with $x \in \mathbb{X}$ for $1 \leq m \leq M$. For $x \in \mathbb{X}$, we then define the optimal foundation representation for $x$ under the $m$-th FRL model as $\boldsymbol{z}_{(m)}^* = f_m^*(x)$ for $1 \leq m \leq M$. By concatenating the optimal representations from all $M$ FRL models, we obtain the fused optimal representations $\boldsymbol{z}^* = (\boldsymbol{z}_{(1)}^{*\top}, \ldots, \boldsymbol{z}_{(M)}^{*\top})^\top$ by directly augmenting the optimal representations from FRL models. By following procedure in Example 1, we denote the optimal candidate representations by $\boldsymbol{z}_{[j]}^*$ for $1 \leq j \leq J$ by applying the candidate construction procedure to optimal foundation representations.

For the downstream task analysis process, for $1 \leq j \leq J$, let $g_j^* = \arg\min_{g \in \mathbb{G}} \mathbb{E}[L(y_i, g(\boldsymbol{z}_{[j]}^*))]$ be the optimal learning function using the $j$-th candidate representation set. Define the risk function of

---

**Algorithm 1** $K$-Fold cross-validation weight criterion for PEARL

---

1: **Input:** Labeled data $D^l = \{(x_i, y_i)\}_{i=1}^n$; Number of folds $K$; $M$ FRL models $\{\widehat{f}_1, \ldots, \widehat{f}_M\}$;
    Surrogate loss function $V$.
2: Construction of the representations: For $i \in \{1, ..., n\}$, obtain $M$ foundation representation sets
    $\widehat{z}_{i,(m)} = \widehat{f}_m(x_i)$ for $1 \le m \le M$ and then $J$ candidate representation sets $\widehat{z}_{i,[j]}$ for $1 \le j \le J$.
3: Cross-validation weight tuning: Partition index set $\mathbb{S} = \{1, \ldots, n\}$ into $K$ folds: $\mathbb{S} = \bigcup_{k=1}^K \mathbb{S}_k$.
4: **for** $j \leftarrow 1$ **to** $J$ **do**;
5:     **for** $k \leftarrow 1$ **to** $K$ **do**
6:         $D_{\text{train}} \leftarrow \{(\widehat{z}_{i,[j]}, y_i)\}_{i \in \mathbb{S} \setminus \mathbb{S}_k}$;
7:         Train the prediction model $\tilde{g}_j^{(-k)}$ on $D_{\text{train}}$;
8:         Obtain $\widetilde{y}_{i,[j]} = \tilde{g}_j^{(-k)}(\widehat{z}_{i,[j]})$ for $i \in \mathbb{S}_k$.
9:     **end for**
10: **end for**
11: The weighted prediction is given by $\widetilde{y}_i(\boldsymbol{w}) = \sum_{j=1}^J w_j \widetilde{y}_{i,[j]}$
12: Weight choice criterion

$$\widehat{\boldsymbol{w}} = \text{argmin}_{\mathbf{w} \in \mathbb{W}} \frac{1}{n} \sum_{i=1}^n V(y_i, \widetilde{y}_i(\boldsymbol{w})).$$

13: **Output:** $\widehat{w}$.

---

model averaging predictor with respect to the label of a new observation $x_{\text{new}}$ by

$$R_L(\boldsymbol{w}) = \mathbb{E}\{L(y_{\text{new}}, \widehat{y}_{\text{new}}(\boldsymbol{w}))\} = \mathbb{E}\{L(y_{\text{new}}, \sum_{j=1}^J w_j \widehat{g}_j(\widehat{z}_{\text{new},[j]}))\}. \tag{5}$$

**Theorem 1.** *Under Assumptions 1-6 in Appendix C, we have*

$$\frac{R_L(\widehat{\boldsymbol{w}})}{\inf_{\mathbf{w} \in \mathbb{W}} R_L(\boldsymbol{w})} \to 1 \tag{6}$$

*in probability, where $\widehat{w}$ is tuned by equation 3.*

Theorem 1 shows the asymptotic optimality of PEARL predictor in terms of downstream risk. It means that PEARL predictor can asymptotically outperform or at least match any FRL or other linear combinations of FRLs. A commonly used linear combination is the simple model averaging with equal weights, $\boldsymbol{w} = (J^{-1}, \ldots, J^{-1})^\top$.

Before we present the weight consistency, we define the correctly specified candidate models that satisfy $g_j^*(z_{\text{new},[j]}^*) = \mathbb{E}(y_{\text{new}}|x_{\text{new}})$. Let $\mathbb{D} \subset \{1, \ldots, J\}$ be a subset of $\{1, \ldots, J\}$ that contains all the index of the correctly specified candidate models. Define $\widehat{\tau} = \sum_{j \in \mathbb{D}} \widehat{w}_j$ be the sum of weights assigned to the these candidate models. Assume that $\mathbb{D}$ is not empty, and it means that at least one correctly specified model is included in the candidate model set, and therefore we only need to prove that $\widehat{\tau} \to 1$ to show the consistency of weights. In addition, since if all candidate models are correctly specified, then the result $\widehat{\tau} = 1$ is directly derived, we assume that $\mathbb{D} \neq \{1, \ldots, J\}$.

**Theorem 2.** *If Assumptions 1-5 and 7-8 in Appendix C are satisfied, then we have $\widehat{\tau} \to 1$.*

Theorem 2 shows that the nonzero weights will asymptotically be put on the correctly specified candidate models when at least one correctly specified model exists in the candidate set. This result implies that, PEARL will asymptotically concentrate on the models whose optimal predicted values coincide with the conditional expectation.

## 4 EXPERIMENTS

This section evaluates PEARL through one simulation study and three real-world datasets. The simulation study uses a nonlinear regression setting, and four real-world datasets consist of CIFAR-10 and CIFAR-100 (Krizhevsky, 2009) for image classification, Polarity (Pang & Lee, 2004) for

sentiment analysis, and Wine Review[1] for multi-modal learning. See Table E.5 in Appendix E for the overall summary of all experiments. For classification tasks, cross-entropy (CE) loss is used as the surrogate function, and squared loss is applied for regression tasks when computing weights.

We compare PEARL with following baseline methods: Best, Fusion, SA-FRL, SA-cand, and MS. Best reports the performance of the single FRL model that achieves the best performance on the downstream task. Fusion refers to a direct concatenation of all FRL representations into one representation vector. SA-FRL is a simple averaging method that equally averages predictions of all individual FRL models. SA-cand is another simple averaging method that equally averages the predictions of all candidate models(differing only in the model set from SA-FRL). MS is a model selection method that selects the best candidate model based on cross-validation.

## 4.1 SYNTHETIC DATASET

This section presents a simulation study on a regression task to evaluate our proposed methods. The data generating process is defined as:

$$y_i = \beta_0 + \beta_1 x_{i1} + \beta_2 x_{i2} + \alpha_1 x_{i1}^2 + \alpha_2 x_{i2}^2 + \gamma x_{i1} x_{i2} + \sin^2(x_{i1}) + \epsilon_i, \tag{7}$$

where observed predictors $\boldsymbol{x}_i = (x_{i1}, x_{i2})^\top$ are independent identically distributed (i.i.d.) from $\mathcal{N}(\boldsymbol{0}, \mathbf{I})$, and $\epsilon_i \sim \mathcal{N}(0, \sigma^2)$. The coefficients $(\beta_0, \beta_1, \beta_2, \alpha_1, \alpha_2, \gamma)$ are drawn independently from $\mathcal{N}(0, 0.09)$. We vary noise levels with $\sigma \in \{0.1, 0.5, 0.9, 1.5\}$ and labeled sample sizes $n \in \{100, 200, 400, 800\}$ for labeled data, fixing unlabeled sample size $N = 2000$. Each configuration is repeated 100 times.

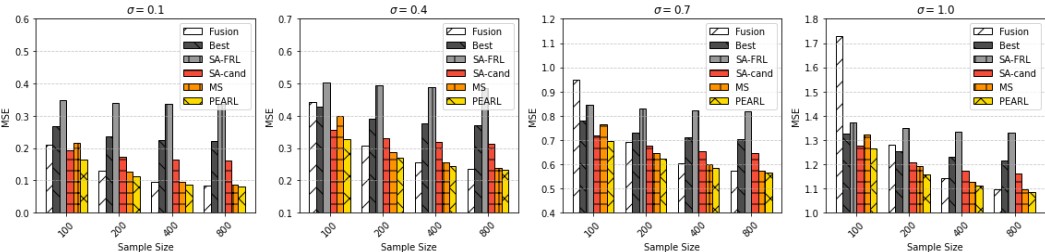

Figure 1: Synthetic-data results

We adopt kernel principal component analysis (KPCA) (Mika et al., 1999) with five kernels as FRL models: linear, Gaussian, polynomial, sigmoid, and cosine kernels. These yield five foundation representation sets, from which we construct $J = 2^5 - 1 = 31$ candidate sets by all non-empty combinations (see Example 1). For each candidate set, a linear regression is fit on the labeled dataset to predict the target variable.

The results in Fig. 1 reveal that our method outperforms the baseline methods in most cases. As the noise level increases, which is proportional to $\sigma$, the performance of the Fusion and MS methods deteriorates more rapidly than that of PEARL, particularly when $n = 100$ or 200. For larger sample sizes ($n = 500$ or 1000), PEARL substantially outperforms Best, SA-FRL and SA-cand. These results demonstrate that PEARL mitigates overfitting in small sample sizes and maximizes predictive performance in larger datasets.

## 4.2 IMAGE CLASSIFICATION ON CIFAR-10 AND CIFAR-100

We evaluate two types of FRL training regimes for image classification on CIFAR-10 and CIFAR-100: (i) FRL representations trained on unlabeled data and (ii) pretrained FRL representations. CIFAR-10 and CIFAR-100 are standard benchmark datasets in image classification. Each dataset consists of 60,000 color images of size $32 \times 32$ pixels with 50,000 training and 10,000 test images. CIFAR-10 contains 10 object categories, while CIFAR-100 contains 100 object categories with a similar image format but a finer-grained label space.

---

[1]https://www.kaggle.com/datasets/zynicide/wine-reviews

### 4.2.1 CIFAR-10 WITH FRL REPRESENTATIONS TRAINED ON UNLABELED DATA

For the CIFAR-10 dataset, we use $M = 4$ FRL models: AE, variational autoencoder (VAE), Sim-CLR (Chen et al., 2020) with a CNN encoder, and SimCLR with a ResNet encoder. For AE and VAE, we also adopt ResNet-based encoders. Based on these $M$ foundation models, we construct $J = 15$ candidate representation sets as in Example 1; implementation details are given in Section Appendix E. For the downstream classification heads, we use either a simple MLP or multinomial logistic regression. We consider labeled sample sizes $n \in \{300, 500, 1500, 3000\}$ and, for each $n$, repeat the experiments 100 times with different random subsets of labeled data, while the FRL models are pretrained once on the full CIFAR-10 training set and reused across all runs. For evaluation, we report three metrics: classification accuracy (Acc), CE, and mean squared error (MSE) between the true and predicted label probabilities. In Table 1 and all subsequent experiments, the best result in each setting is highlighted in bold and the second-best is underlined.

As shown in Table 1, PEARL consistently outperforms baselines across all metrics and sample sizes, demonstrating superior classification accuracy and better calibration. Compared with MS, which successfully selects the single FRL model with the highest performance since MS has the same performance with Best, PEARL further improves the results. This demonstrates the clear advantage of combining different models rather than relying on a single one. Moreover, when compared with SA-FRL and SA-cand, PEARL also achieves superior performance. This highlights that different models contribute unequally to the downstream task, and assigning task-specific weights is more effective than applying equal weights. Finally, relative to Fusion, the weighted averaging strategy in PEARL yields better results. This confirms that learning to weight different representation groups is more beneficial than simply concatenating them.

Table 1: Results of different methods on CIFAR-10 dataset

|  | n | Fusion | Best | SA-FRL | SA-cand | MS | PEARL |
|---|---|---|---|---|---|---|---|
| Acc(%) | 300 | 42.129 (1.938) | 53.379 (0.836) | 42.125 (2.441) | 43.839 (1.992) | 53.379 (0.836) | **53.825 (0.831)** |
|  | 600 | 48.748 (1.600) | 56.133 (0.529) | 48.449 (1.416) | 50.791 (1.527) | 56.133 (0.529) | **57.028 (0.521)** |
|  | 1500 | 56.389 (0.706) | 58.432 (0.380) | 53.726 (0.775) | 57.663 (0.701) | 58.432 (0.380) | **60.186 (0.375)** |
|  | 3000 | 60.342 (0.456) | 59.518 (0.251) | 55.791 (0.493) | 60.704 (0.456) | 59.817 (0.566) | **61.928 (0.356)** |
| CE | 300 | 2.589 (0.303) | 1.292 (0.023) | 1.587 (0.031) | 1.614 (0.075) | 1.292 (0.023) | **1.282 (0.023)** |
|  | 600 | 1.983 (0.220) | 1.212 (0.011) | 1.475 (0.013) | 1.399 (0.041) | 1.212 (0.011) | **1.190 (0.012)** |
|  | 1500 | 1.350 (0.047) | 1.149 (0.006) | 1.404 (0.006) | 1.233 (0.013) | 1.149 (0.006) | **1.104 (0.007)** |
|  | 3000 | 1.140 (0.015) | 1.120 (0.004) | 1.378 (0.005) | 1.173 (0.007) | 1.114 (0.010) | **1.054 (0.006)** |
| MSE | 300 | 0.087 (0.005) | 0.060 (0.001) | 0.071 (0.002) | 0.072 (0.003) | 0.060 (0.001) | **0.059 (0.001)** |
|  | 600 | 0.075 (0.004) | 0.057 (0.000) | 0.067 (0.001) | 0.064 (0.002) | 0.057 (0.000) | **0.056 (0.000)** |
|  | 1500 | 0.060 (0.001) | 0.054 (0.000) | 0.064 (0.000) | 0.057 (0.001) | 0.054 (0.000) | **0.052 (0.000)** |
|  | 3000 | 0.053 (0.001) | 0.053 (0.000) | 0.063 (0.000) | 0.054 (0.000) | 0.053 (0.001) | **0.050 (0.000)** |

### 4.2.2 CIFAR-10 AND CIFAR-100 WITH PRETRAINED FRL REPRESENTATIONS

To assess whether PEARL continues to yield performance gains when built on top of stronger FRL models, we further consider FRLs based on pretrained CNN and transformer backbones on CIFAR-10 and CIFAR-100. The specific backbone architectures are detailed in Appendix E.

On CIFAR-10 and CIFAR-100, we evaluate PEARL and all competing methods under a common protocol with $n \in \{1500, 3000\}$ labeled samples. For CIFAR-10, we set $M = 4$ and use four pretrained ResNet-type models as FRLs; for CIFAR-100, we use $M = 4$ pretrained transformer-based FRLs with ViT/Swin backbones. In both cases, the induced representations are used to construct candidate representation sets exactly as in the base configuration.

Table 2 reports classification accuracy (Acc) and cross-entropy loss (CE) for CIFAR-10, while Table 3 reports Acc and CE for CIFAR-100 (MSE is omitted for brevity). Across both datasets and label budgets, PEARL consistently outperforms all baselines, with particularly notable reductions in CE, indicating better-calibrated predictions. These results show that the improvements of PEARL are not an artifact of a particular FRL configuration: they persist when PEARL is built on stronger ResNet/ViT-based representations, and PEARL scales well to larger and more challenging image-classification tasks.

Table 2: Results of pretrained FRLs on CIFAR-10 dataset

| | n | Fusion | Best | SA-FRL | SA-cand | MS | PEARL |
|---|---|---|---|---|---|---|---|
| Acc(%) | 1500 | 88.953 (0.301) | 90.371 (0.271) | 90.431 (0.277) | 89.789 (0.281) | 90.463 (0.280) | **91.340 (0.231)** |
| | 3000 | 90.345 (0.228) | 91.095 (0.220) | 91.557 (0.200) | 91.094 (0.202) | 91.583 (0.250) | **92.316 (0.217)** |
| CE | 1500 | 0.362 (0.011) | 0.294 (0.007) | 0.307 (0.005) | 0.295 (0.006) | 0.289 (0.008) | **0.259 (0.006)** |
| | 3000 | 0.333 (0.009) | 0.280 (0.006) | 0.275 (0.004) | 0.259 (0.004) | 0.262 (0.007) | **0.231 (0.004)** |

Table 3: Results of pretrained FRLs on CIFAR-100 dataset

| | n | Fusion | Best | SA-FRL | SA-cand | MS | PEARL |
|---|---|---|---|---|---|---|---|
| Acc(%) | 1500 | 84.013 (0.547) | 84.936 (0.394) | 83.572 (0.534) | 84.364 (0.489) | 85.688 (0.434) | **86.241 (0.411)** |
| | 3000 | 87.178 (0.303) | 87.234 (0.283) | 86.498 (0.281) | 87.519 (0.286) | 87.937 (0.250) | **88.629 (0.260)** |
| CE | 1500 | 0.579 (0.021) | 0.575 (0.020) | 0.666 (0.015) | 0.599 (0.015) | 0.531 (0.018) | **0.505 (0.016)** |
| | 3000 | 0.460 (0.011) | 0.482 (0.010) | 0.529 (0.007) | 0.462 (0.007) | 0.439 (0.010) | **0.396 (0.009)** |

## 4.3 POLARITY DATASET

We use the Polarity v2.0 dataset (Pang & Lee, 2004), a standard benchmark for sentiment analysis containing 2000 movie reviews evenly split between positive and negative labels. The task is binary sentiment classification. We consider $M = 6$ FRL models, including CountVectorizer, TF-IDF (term frequency-inverse document frequency), Doc2Vec (Le & Mikolov, 2014), Word2Vec (Mikolov et al., 2013), BERT, and GloVe (Pennington et al., 2014). Among these, BERT and GloVe are pre-trained models, while the others are trained from scratch. For downstream classification, we apply either a simple MLP or logistic regression on the learned representations, using a 70%-30% train–test split repeated 10 times for robustness.

Table 4: Results of different methods on Polarity dataset

| | Fusion | Best | SA | PEARL |
|---|---|---|---|---|
| Acc(%) | 83.587 (1.445) | 87.745 (1.113) | 86.233 (1.392) | **88.873 (1.173)** |
| CE | 0.482 (0.045) | 0.308 (0.014) | 0.423 (0.009) | **0.291 (0.013)** |
| MSE | 0.126 (0.010) | 0.092 (0.005) | 0.128 (0.004) | **0.086 (0.005)** |

Given the high dimensionality of CountVectorizer and TF-IDF, no additional candidate sets are constructed, yielding $M = J = 6$. In this case, SA-cand reduces to SA-FRL, and MS is dominated by Best, so both are omitted from comparison. As shown in Table 4, PEARL consistently outperforms the remaining baselines, demonstrating its effectiveness even without additional candidate sets.

To further illustrate the applicability of PEARL to large language model (LLM)-based representations, we extend the sentiment classification experiment by using pretrained LLMs as FRLs. Specifically, we consider three LLMs: `Qwen/Qwen2.5-1.5B-Instruct`, `meta-llama/Llama-3.2-1B-Instruct`, and `openai-community/gpt2-xl`. The first two are open-source instruction-tuned LLMs, and the third is a GPT-series model available via the `openai-community` repository. For each LLM, we extract fixed sentence-level representations for each review and build a candidate model on top of its representation. See Appendix E for implementation details. Based on these three representations, we further construct $J = 2^3 - 1 = 7$ candidate models using all non-empty subsets of the three representation sets. PEARL and the competing methods are then applied to these $J$ candidates.

From Table 5, we see that PEARL consistently outperforms all baselines in terms of Acc, CE, and MSE, achieving the highest accuracy (96.716%) and the lowest CE (0.091) among all methods. These results show that our framework effectively combines multiple pretrained LLM representations for the target task with theoretical guarantees. At the same time, we observe that the single best LLM-based FRL (Best) only achieves 92.95% accuracy and a CE of 0.209, further underscoring the limitations of relying on a single model and the importance of aggregating representations.

Table 5: Polarity dataset with pretrained LLM-based FRLs.

| | Fusion | Best | SA-rep | SA | MS | PEARL |
|---|---|---|---|---|---|---|
| Acc(%) | 94.827 (0.747) | 92.950 (0.892) | 95.343 (0.691) | 95.488 (0.705) | 96.407 (0.645) | **96.716 (0.645)** |
| CE | 0.139 (0.019) | 0.209 (0.024) | 0.165 (0.007) | 0.130 (0.007) | 0.100 (0.015) | **0.091 (0.011)** |
| MSE | 0.039 (0.005) | 0.055 (0.006) | 0.046 (0.003) | 0.035 (0.003) | 0.027 (0.004) | **0.026 (0.004)** |

## 4.4 Wine Review Dataset

We evaluate PEARL in a multi-modal setting using the Wine Review dataset (approximately 130k records), where each review contains unstructured text and structured attributes (variety, location, winery, and price). The task is to predict the wine quality score as a continuous outcome. For representation learning, we use PCA and VAEs (Kingma & Welling, 2014) for numerical attributes, and pre-trained BERT and GPT-2 (Radford et al., 2019) for text, trained on 10k unlabeled samples. Linear regression is applied to labeled sets of size $n \in \{100, 200, 500, 1000, 2000\}$, with evaluation on 5k test samples. Each split is repeated 10 times, and performance is assessed by average MSE.

Table 6: Results of different methods on wine review dataset

| n | Best | Fusion | SA-repr | SA-cand | MS | PEARL |
|---|---|---|---|---|---|---|
| 100 | 7.296 (0.588) | 6.159 (0.537) | 26.849 (56.176) | 17.670 (28.147) | 6.468 (0.592) | **5.726 (0.384)** |
| 200 | 7.596 (0.451) | 5.678 (0.353) | 5.251 (0.291) | **4.973 (0.274)** | 6.203 (0.792) | 5.103 (0.334) |
| 500 | 7.124 (0.335) | 6.497 (0.350) | 5.441 (0.264) | 5.371 (0.268) | 6.557 (0.393) | **4.920 (0.193)** |
| 1000 | 6.730 (0.264) | 10.336 (0.659) | 5.407 (0.206) | 6.171 (0.286) | 6.878 (0.550) | **5.332 (0.247)** |
| 2000 | 5.496 (0.136) | 14.210 (1.002) | 4.147 (0.131) | 4.416 (0.127) | 5.043 (0.299) | **3.659 (0.084)** |

From Table 6, across most sample sizes, PEARL achieves the lowest MSE compared with both single-modality baselines (Best) and other averaging strategies, showing clear advantages in integrating heterogeneous representations from multi-modal datasets. These results underscore that PEARL not only improves prediction accuracy but also maintains robustness in the challenging multi-modal setting, highlighting its strong generalizability beyond unimodal tasks.

## 5 Conclusion

In this work, we introduce a theoretically grounded aggregation method for representation learning based on frequentist model averaging. The proposed framework is widely applicable and highly general, accommodating tasks ranging from matrix/tensor decomposition to nonlinear representation learning in deep neural networks. This broad scope underscores its ability to bridge interpretable statistical tools with modern AI techniques, making it suitable for diverse real-world applications. A natural next step is to move beyond prediction risk and develop valid confidence intervals and hypothesis tests, thereby strengthening the inferential foundations of model averaging with representations.

## Use of Large Language Models

The authors used ChatGPT to polish the language by checking grammar, correcting minor errors, and suggesting improvements in clarity.

## Reproducibility Statement

We will publicly release the complete codebase for all methods and experiments in an open-source repository upon acceptance, with implementation and experimental settings detailed in Appendix E.

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
