## APPENDIX A    PEARL FOR MULTIMODAL DATA

This part illustrates how the proposed method PEARL is applied to multimodal representation learning. In many practical applications, data are available in multiple modalities—such as images and text for news articles or sequences of images, audio, and text for YouTube videos. The proposed method PEARL can effectively combine representations learned from each modality.

Suppose our predictor consists of $S$ distinct modalities, and the predictor space in the $s$-th modality is defined by $\mathbb{X}^s$. Define the overall predictor space by

$$\mathbb{X} = \mathbb{X}^1 \cup \mathbb{X}^2 \cup \cdots \cup \mathbb{X}^S,$$

where the union symbol is used in an abstract sense[A.1]. We consider a large collection of $N$ unlabeled samples, $\mathcal{D}^u = \{x'_i \in \mathbb{X} : i = 1, \ldots, N\}$, to train a total of $M$ FRL models. These FRL models are designed to learn representations from the multimodal input data, and they can operate on single modalities or combinations thereof. For illustration, consider two example FRL models for the input $x = (x^1, \ldots, x^S) \in \mathbb{X}$:

$$f_1 : \mathbb{X}^1 \to \mathbb{R}^{p_1}, \quad x^1 \mapsto \boldsymbol{z}_{(1)},$$
$$f_2 : (\mathbb{X}^2, \mathbb{X}^3) \to \mathbb{R}^{p_2}, \quad (x^2, x^3) \mapsto \boldsymbol{z}_{(2)}.$$

Here, $f_1$ represents FRL models that focus on individual modalities, while $f_2$ represents those built to capture joint information from a combination of modalities (see, e.g., (Ngiam et al., 2011; Shekhar et al., 2014; Gu et al., 2017)). The complete set of trained FRL models is denoted as $\{\widehat{f}_1, \ldots, \widehat{f}_M\}$.

For the downstream task, we assume that there are $n$ labeled samples forming the dataset $\mathcal{D}^l = \{(x_i, y_i) \in \mathbb{X} \times \mathcal{Y} : i = 1, \ldots, n\}$ where each $x_i = (x_i^1, x_i^2, \ldots, x_i^S)$ is a multimodal input and $y_i$ is the corresponding label. For example, the foundational representations for sample $x_i$ obtained via the first two FRL models are $\widehat{\boldsymbol{z}}_{i,(1)} = \widehat{f}_1(x_i^1)$ and $\widehat{\boldsymbol{z}}_{i,(2)} = \widehat{f}_2(x_i^2, x_i^3)$, respectively. After extracting representations from all $M$ FRL models, we generate $J$ candidate representation sets, denoted by $\widehat{\boldsymbol{z}}_{i,[j]}$ for $i = 1, \ldots, n$ and $j = 1, \ldots, J$. The subsequent steps of our methodology follow the procedure outlined in our main algorithm, where we derive cross-validation weights, fit the prediction models, and make prediction for any new observation $x_{\text{new}} \in \mathbb{X}$.

## APPENDIX B    ADDITIONAL EMPIRICAL STUDIES

### B.1    COMPUTATIONAL COST

We report the runtime of PEARL on CIFAR-10 and the Polarity dataset using two types of downstream models $g$: logistic regression and a small MLP. For each setting, we report the runtime of (i) CV ($K = 10$, 10 CPU cores), (ii) downstream task training time ($\widehat{g}_j$), and (iii) weight-optimization time. As shown below, the overall runtimes including CV runtime remain small.

Table B.1: Runtime comparison across datasets and downstream model types

| Dataset | $M$ | $J$ | Type of $g$ | CV (parallel) [s] | Downstream Training [s] | Weight Optimization [s] | Total Runtime [s] |
|---|---|---|---|---|---|---|---|
| CIFAR-10 | 4 | 15 | Logistic | 4.953 | 5.195 | 0.129 | 10.277 |
|  |  |  | MLP | 7.680 | 12.891 | 0.152 | 20.723 |
| Polarity | 6 | 6 | Logistic | 7.421 | 2.541 | 0.004 | 9.966 |
|  |  |  | MLP | 8.586 | 4.471 | 1.169 | 14.226 |

We also compare the runtime of one-time weight optimization of PEARL with linear weighting and nonlinear weighting on CIFAR-10 dataset. For nonlinear weighting, we take neural network (NN) aggregator as example. Specifically, let $\hat{g}_j(x)$ denote the prediction vector of the $j$-th candidate model on input $x$. For each training sample $x_i$, we form the concatenated vector

---

[A.1]In practice, the integration of heterogeneous source data might involve taking a Cartesian product or another form of data fusion, depending on how the features interact. Therefore, $\cup$ can be possibly replaced by a more specific product or joint space.

$v_i = \big(\hat{g}_1(x_i), \ldots, \hat{g}_J(x_i)\big)$ and train an two layer MLP $h_\theta$ to predict $y_i$ from $v_i$. From the table below, we can see that linear weighting optimization is much faster than NN aggregator optimization.

Table B.2: Runtime comparison between linear weighting and nonlinear weighting

| Dataset | Linear weighting (s) | NN aggregator (s) |
|---------|---------------------|-------------------|
| CIFAR-10 | 0.129 | 2.235 |

## B.2 COMPARISON WITH A NONLINEAR NEURAL AGGREGATION BASELINE

We conduct experiments incorporating a nonlinear neural network (NN) aggregator. Let $\hat{g}_j(x)$ denote the prediction vector of the $j$-th candidate model on input $x$. For each training sample $x_i$, we form the concatenated vector

$$v_i = \big(\hat{g}_1(x_i), \ldots, \hat{g}_J(x_i)\big),$$

and train a multilayer perceptron (MLP) $h_\theta$ to predict $y_i$ from $v_i$. This nonlinear baseline is referred to as *NN-aggregator*. We evaluated NN-aggregator on CIFAR-10 and the Polarity dataset under the same candidate models as PEARL. The results are summarized in Table B.3.

Table B.3: Comparison between PEARL and a nonlinear neural network aggregator.

| Dataset | Metric | $n$ | PEARL | NN-aggregator |
|---------|--------|-----|-------|---------------|
| CIFAR-10 | Acc(%) | 300 | 53.83 (0.83) | 44.81 (1.24) |
| | | 600 | 57.03 (0.52) | 50.83 (0.97) |
| | | 1500 | 60.19 (0.38) | 56.56 (0.62) |
| | | 3000 | 61.93 (0.36) | 60.25 (0.53) |
| | CE | 300 | 1.282 (0.023) | 3.331 (0.174) |
| | | 600 | 1.190 (0.012) | 2.754 (0.178) |
| | | 1500 | 1.104 (0.007) | 1.674 (0.070) |
| | | 3000 | 1.054 (0.006) | 1.293 (0.023) |
| Polarity | Acc(%) | 1400 | 88.873 (1.173) | 86.029 (1.264) |
| | CE | 1400 | 0.291 (0.013) | 0.358 (0.039) |

Across both datasets and all sample sizes, the nonlinear NN-aggregator consistently underperforms PEARL in terms of both accuracy and cross-entropy. In addition, PEARL relies on a linear weighting scheme, which provides transparent and interpretable importance weights over candidate models, whereas the NN-based ensemble operates as a black box without comparable interpretability or theoretical guarantees.

## B.3 ADDITIONAL REAL DATA: MNIST DATASET

We also evaluate PEARL on the MNIST benchmark for image classification. We use similar configuration for FRL methods, except that the ResNet encoders are replaced with CNN ones. For downstream classification, we employ either a simple MLP or multinomial logistic regression. As shown in Table B.4, PEARL consistently outperforms baselines in MNIST dataset. The implementation details are provided in Appendix E.

## APPENDIX C    THEORETICAL ASSUMPTIONS AND PROOFS OF THEOREMS

Let a function $h^*(x, \boldsymbol{w}) = \sum_{j=1}^{J} w_j g_j^*(\boldsymbol{z}_{[j]}^*)$ be the model averaging value of outputs derived from optimal candidate representations and optimal downstream task learning function, as $z_{[j]}^*$ depends on $x$. Define $\boldsymbol{w}^* = (w_1, \ldots, w_J)^\top$ that satisfies $\boldsymbol{w}^* = \arg\min_{\mathbf{w} \in \mathbb{W}} \mathbb{E}[L(y, h^*(x, \boldsymbol{w}))]$. Analogously, for the estimated values, define $\widehat{h}(x, \boldsymbol{w}) = \sum_{j=1}^{J} w_j \widehat{g}_j(\widehat{\boldsymbol{z}}_{[j]})$.

Table B.4: Results of different methods on MNIST dataset

|        | n    | Fusion        | Best          | SA-FRL        | SA-cand       | MS              | PEARL           |
|--------|------|---------------|---------------|---------------|---------------|-----------------|-----------------|
| Acc(%) | 300  | 94.647 (0.509) | 95.390 (0.385) | 94.140 (0.501) | 94.863 (0.467) | 96.242 (0.478)  | **96.355 (0.451)** |
|        | 600  | 95.949 (0.309) | 96.016 (0.236) | 95.064 (0.295) | 96.062 (0.282) | 96.984 (0.286)  | **97.086 (0.237)** |
|        | 1500 | 97.066 (0.204) | 96.879 (0.165) | 96.095 (0.191) | 97.241 (0.193) | 97.625 (0.187)  | **97.762 (0.159)** |
|        | 3000 | 97.570 (0.143) | 97.316 (0.145) | 96.578 (0.133) | 97.844 (0.120) | 97.939 (0.165)  | **98.090 (0.122)** |
| CE     | 300  | 0.170 (0.016) | 0.152 (0.015) | 0.238 (0.011) | 0.178 (0.011) | 0.122 (0.015)   | **0.121 (0.013)** |
|        | 600  | 0.132 (0.011) | 0.127 (0.007) | 0.196 (0.006) | 0.140 (0.006) | 0.096 (0.009)   | **0.094 (0.006)** |
|        | 1500 | 0.099 (0.008) | 0.099 (0.005) | 0.161 (0.003) | 0.106 (0.004) | 0.074 (0.006)   | **0.071 (0.004)** |
|        | 3000 | 0.082 (0.006) | 0.085 (0.003) | 0.146 (0.002) | 0.090 (0.002) | 0.064 (0.005)   | **0.059 (0.003)** |
| MSE    | 300  | 0.008 (0.001) | 0.007 (0.001) | 0.011 (0.001) | 0.008 (0.001) | 0.006 (0.001)   | **0.006 (0.001)** |
|        | 600  | 0.006 (0.000) | 0.006 (0.000) | 0.009 (0.000) | 0.006 (0.000) | 0.005 (0.000)   | **0.004 (0.000)** |
|        | 1500 | 0.004 (0.000) | 0.005 (0.000) | 0.007 (0.000) | 0.005 (0.000) | 0.004 (0.000)   | **0.003 (0.000)** |
|        | 3000 | 0.004 (0.000) | 0.004 (0.000) | 0.007 (0.000) | 0.004 (0.000) | 0.003 (0.000)   | **0.003 (0.000)** |

**Assumption C.1.** *Suppose that there exist constants $\mu > 0$ and $c_1 > 0$ such that when $\mathbb{E}[V(y, \widehat{h}(x, \boldsymbol{w}))] - \mathbb{E}[V(y, h^*(x, \boldsymbol{w}^*))] \leq \epsilon$ for $\boldsymbol{w} \in \mathbb{W}$ and a small error $\epsilon > 0$, then*

$$\mathbb{E}[L(y, \widehat{h}(x, \boldsymbol{w}))] - \mathbb{E}\left[L(y, h^*(x, \boldsymbol{w}^*))\right] \leq c_1 \epsilon^\mu.$$

Assumption C.1 requires that a small bias bound in the surrogate function $V$ can translate into a relatively small bias bound in the true loss function $L$. The polynomial relationship between the bias bound of $V$ and $L$ is also adapted in Assumption A1 in Dai et al. (2022). Such conditions hold in many standard machine-learning problems where $V$ is carefully designed to approximate $L$, such that by controlling the bias of the surrogate loss, we also control the bias of the true loss.

**Assumption C.2.** *For each $1 \leq i \leq n$ and $\boldsymbol{w} \in \mathbb{W}$, if $\mathbb{E}[V(y_i, \widetilde{y}_i(\boldsymbol{w})) - V(y_i, y_i^*(\boldsymbol{w}))] \leq \epsilon$ for some small error $\epsilon > 0$, then there exist constants $\kappa > 0$ and $c_2 > 0$ such that,*

$$\mathrm{Var}[V(y_i, \widetilde{y}_i(\boldsymbol{w})) - V(y_i, y_i^*(\boldsymbol{w}))] \leq c_2 \epsilon^\kappa.$$

Assumption C.2 bounds the variance of the loss difference between $\widetilde{y}_i(\boldsymbol{w})$ and $y_i^*(\boldsymbol{w})$ if the expected difference is small in the surrogate function $V$. This assumption requires a mild smoothness for $V$, where the constant $\kappa$ reflects the smoothness. Assumption C.2 is similar to Assumption A2 in Dai et al. (2022).

**Assumption C.3.** *For any $\boldsymbol{w} \in \mathbb{W}$, $\mathrm{Var}(V(y, h^*(x, \boldsymbol{w})) \leq c_3$ with a positive constant $c_3$.*

Assumption C.3 requires that variances of the surrogate loss of the model averaging estimator derived from optimal representations and the downstream task learning function be bounded by a positive constant. This indicates that the data distribution and the model do not produce arbitrarily large fluctuations in the surrogate loss.

**Assumption C.4.** *For two predicted outputs $y'$ and $y''$, it has $|V(y, y') - V(y, y'')| \leq K(y)\|y' - y''\|$, where $K(y)$ is independent of $y'$ and $y''$, and satisfies $\mathbb{E}^{1/2}(|K(y)|^2) \leq c_4$ for some positive constant $c_4$.*

Assumption C.4 requires the smoothness of the surrogate function, and it shows that small changes in predictions lead to small variations in the surrogate loss. It is similar to Assumption 2 in Yu et al. (2025), which poses the stochastic equicontinuity on the measure of divergence in the original loss function, and Assumption A in Zhu et al. (2016) and Assumption 4.2 in Bi et al. (2018), the two of which pose the smoothness on the likelihood function derived the original loss function. Comparing with these smoothness assumptions directly on $L$, we instead apply the assumption to $V$. This broadens the range of admissible original loss functions, allowing those that are not smooth themselves but for which a suitably chosen $V$ can satisfy the smoothness requirement.

Assumption C.4 can be satisfied when the surrogate loss is a *margin-based* loss. For hinge loss $V(y, f) = \max\{0, \ 1 - yy'\}$ under a binary classification setting with $y \in \{-1, +1\}$, it satisfies that

$$\left|\max\{0, \ 1 - yy'\} - \max\{0, \ 1 - yy''\}\right| \ \leq \ \left|(1 - yy') - (1 - yy'')\right| \ = \ |y||y' - y''| \ \leq \ |y' - y''|,$$

where $K(y) = 1$. For the logistic loss (cross-entropy loss) $V(y, y') = \log(1 + e^{-yy'})$ with $y \in \{-1, +1\}$, its derivative with respect to $y'$ satisfies

$$\frac{\partial}{\partial y'} \log(1 + e^{-yy'}) = -\frac{ye^{-yy'}}{1 + e^{-yy'}},$$

whose absolute value is bounded by 1. Hence, the logistic loss is Lipschitz in $y'$ with a constant at most $|y|$ and $K(y) = 1$ in Assumption C.4. More generally, many margin-based losses take the form $V(y, y') = \phi(yy')$ for some convex and often smooth function $\phi$, including the exponential loss, the smoothed hinge loss, and various robust surrogates. In each case, one typically finds a bounded derivative with respect to $y'$, thus admitting a finite Lipschitz constant $K(y)$ that depends at most on $|y|$. For binary classification with $y \in \{-1, +1\}$, we immediately get $|y| = 1$, yielding a constant $K(y) \leq 1$. In multi-class or other extended settings, one can assume $\|y\| \leq C_y$ with high probability, ensuring $\mathbb{E}[K(y)^2] < c_4$.

**Assumption C.5.** *There exists a sequence $\alpha_{N,n}$ depending on $N$ and $n$, such that*

$$\max_{1 \leq j \leq J} \mathbb{E}^{1/2}(\|\widehat{g}_j(\widehat{z}_{[j]}) - g_j^*(z_{[j]}^*)\|^2) = O(\alpha_{N,n})$$

*and*

$$\max_{1 \leq j \leq J} \mathbb{E}^{1/2}(\|\widetilde{g}_j(\widehat{z}_{[j]}) - g_j^*(z_{[j]}^*)\|^2) = O(\alpha_{N,n}).$$

Assumption C.5 shows the maximum error bound of $J$ candidate estimators and their corresponding CV estimators to the limiting values for $J$ candidate models. In practice, $\alpha_{N,n}$ may shrink to 0 as $N, n \to \infty$, reflecting the convergence rate of estimators and their corresponding CV estimators. The requirement holds over $j = 1, \ldots, J$, which is natural in model-averaging contexts where multiple candidate models are considered.

The error bound of $\widehat{g}_j(\widehat{z}_{i,[j]})$ can be decomposed by the approximation error (the discrepancy introduced by training downstream learning function) and the estimation error (the discrepancy introduced by estimating representations) as follows

$$\|\widehat{g}_j(\widehat{z}_{i,[j]}) - g_j^*(z_{i,[j]}^*)\| \leq \underbrace{\|\widehat{g}_j(z_{i,[j]}^*) - g_j^*(z_{i,[j]}^*)\|}_{\text{approximation error}} + \underbrace{\|\widehat{g}_j(\widehat{z}_{i,[j]}) - \widehat{g}_j(z_{i,[j]}^*)\|}_{\text{estimation error}}. \tag{C.1}$$

The approximation error depicts the ability of model $\widehat{g}_j$ in downstream task analysis step to approximate the optimal model $g_j^*$. The approximation error can potentially be reduced by using more flexible model specifications, which allow experts to better tailor the model to the data. However, increasing model flexibility also introduces the risk of overfitting. Overfitting occurs when the model becomes too closely aligned with the specific patterns in the training data, which may lead to poor performance when applied to new and unseen data. The estimation error arises due to the representation learning process. If the used function $\widehat{g}$ falls in the Lipschitz-continuous function space, then the estimation error is bounded by $c^L \|\widehat{z}_{i,[j]} - z_{i,[j]}^*\|$ with a positive Lipschitz constant $c^L$. Analogously, the estimation error of $\widetilde{g}_j(\widehat{z}_{i,[j]})$ can be decomposed similar to equation C.1 and it is assumed that the cross validated estimator inherits the convergence rate of the original estimator.

In addition, the framework can be extended to encompass commonly used pretrained models as foundation models. Unlike the assumption in the original formulation, where all foundation models are trained on the same unlabeled dataset, these pretrained models may be trained on different unsupervised datasets. Subsequently, they use the learned representations for downstream task analysis. Specifically, let $\mathcal{D}_m^u$ denote the unsupervised dataset for the $m$-th foundation model with a sample size of $N_m$. To generalize the framework for these pretrained models, we modify Assumption C.5 by setting $N = \max_{1 \leq m \leq M} N_m$.

## C.1 ASSUMPTIONS AND PROOF OF THEOREM 1

### C.1.1 ASSUMPTIONS

Now, let's define the risk function of the weighted predictor using the limiting values of each model's prediction:

$$R_L^*(\boldsymbol{w}) = \mathbb{E}\{L(y_{\text{new}}, y_{\text{new}}^*(\boldsymbol{w}))\} = \mathbb{E}\Big\{L\Big(y_{\text{new}}, \sum_{j=1}^J \omega_j g_j^*(z_{\text{new},[j]}^*)\Big)\Big\}. \tag{C.2}$$

We define the surrogate risk function $R_V(\boldsymbol{w})$ of our model averaging estimator by replacing $L$ with $V$ in equation 5. The corresponding risk function with limiting values $R_V^*(\boldsymbol{w})$ is then defined by replacing $L$ with $V$ in equation C.2. Define the infimum risk of $R_L^*(\boldsymbol{w})$ by $\xi_L = \inf_{\mathbf{w}\in\mathbb{W}} R_L^*(\boldsymbol{w})$, and the infimum risk of $R_V^*(\boldsymbol{w})$ by $\xi_V = \inf_{\mathbf{w}\in\mathbb{W}} R_V^*(\boldsymbol{w})$.

**Assumption C.6.** $\xi_V^{-1}\xi_L = O(1)$ *and* $\xi_V^{-1}\max\{\alpha_{N,n}^{\kappa/2} n^{-1/2}, \alpha_{N,n}^\kappa, \alpha_{N,n}^\mu\} = o(1)$.

Assumption C.6 contains two parts. Part $I = \xi_V^{-1}\xi_L = O(1)$ keeps minimal possible surrogate risk on the same order with the minimal possible true loss. It is a mild or non-degenerate condition ensuring that improvements on the surrogate side can be translated to improvements on the true loss. To put it in another words, if $\xi_V$ is extremely larger than $\xi_L$, it may imply that even the best surrogate solution performs poor in the true loss, making it pointless to minimize $V$ instead of $L$. Part $II = \xi_V^{-1}\max\{\alpha_{N,n}^{\kappa/2} n^{-1/2}, \alpha_{N,n}^\kappa, \alpha_{N,n}^\mu\} = o(1)$ means that the minimal possible surrogate risk $\xi_V$ diverges at the rate faster than $\max\{\alpha_{N,n}^{\kappa/2} n^{-1/2}, \alpha_{N,n}^\kappa, \alpha_{N,n}^\mu\}$. The term $\alpha_{N,n}$ measures the approximation error and estimation error of the two stages, and the term $\alpha_{N,n}^{\kappa/2} n^{-1/2}$ arises by controlling the cross-validation and sample-average difference under polynomial variance bounds, and $\alpha_{N,n}^\mu$ arises by converting the risk difference of surrogate loss to that of the true loss.

### C.1.2 PROOF OF THEOREM 1

*Proof.* We know that

$$R_L(\widehat{\boldsymbol{w}}) - \inf_{\mathbf{w}\in\mathbb{W}} R_L(\boldsymbol{w}) = \Big(R_L(\widehat{\boldsymbol{w}}) - \inf_{\mathbf{w}\in\mathbb{W}} R_L^*(\boldsymbol{w})\Big) + \Big(\inf_{\mathbf{w}\in\mathbb{W}} R_L^*(\boldsymbol{w}) - \inf_{\mathbf{w}\in\mathbb{W}} R_L(\boldsymbol{w})\Big), \tag{C.3}$$

and we denote $\mathrm{I} = R_L(\widehat{\boldsymbol{w}}) - \inf_{\mathbf{w}\in\mathbb{W}} R_L^*(\boldsymbol{w})$ and $\mathrm{II} = \inf_{\mathbf{w}\in\mathbb{W}} R_L^*(\boldsymbol{w}) - \inf_{\mathbf{w}\in\mathbb{W}} R_L(\boldsymbol{w})$.

For I, we know $\inf_{w\in\mathbb{W}} R_L^*(\boldsymbol{w}) = \mathbb{E}[L(y_{\text{new}}, h^*(x_{\text{new}}, \boldsymbol{w}^*))]$. Based on Assumption C.1, we need to bound the difference in $V$ risk that leads to a bound in $L$ risk. We decompose $R_V(\widehat{\boldsymbol{w}}) - \mathbb{E}[V(y_{\text{new}}, h^*(x_{\text{new}}, \boldsymbol{w}^*))]$ by

$$R_V(\widehat{\boldsymbol{w}}) - \mathbb{E}[V(y_{\text{new}}, h^*(x_{\text{new}}, \boldsymbol{w}^*))]$$

$$= \mathbb{E}[V(y_{\text{new}}, h^*(x_{\text{new}}, \boldsymbol{w}^*))] \left\{ \frac{R_V(\widehat{\boldsymbol{w}})}{\inf_{\mathbf{w}\in\mathbb{W}} R_V(\boldsymbol{w})} \times \frac{\inf_{\mathbf{w}\in\mathbb{W}} R_V(\boldsymbol{w})}{\mathbb{E}[V(y_{\text{new}}, h^*(x_{\text{new}}, \boldsymbol{w}^*))]} - 1 \right\}.$$

Under Assumption C.4 and Assumption C.5, we know that

$$\sup_{\mathbf{w}\in\mathbb{W}} \frac{|R_V(\boldsymbol{w}) - R_V^*(\boldsymbol{w})|}{R_V^*(\boldsymbol{w})}$$

$$\leq \xi_V^{-1} \sup_{\mathbf{w}\in\mathbb{W}} \mathbb{E}\left\{ V\Big(y_{\text{new}}, \sum_{j=1}^J \omega_j \widehat{g}_{[j]}(\widehat{\boldsymbol{z}}_{\text{new},[j]})\Big) - V\Big(y_{\text{new}}, \sum_{j=1}^J \omega_j g_{[j]}^*(z_{\text{new},[j]}^*)\Big) \right\}$$

$$\leq \xi_V^{-1} \mathbb{E}^{1/2}(\|y_{\text{new}}\|^2) \max_{1\leq j\leq J} \mathbb{E}^{1/2}(\|\widehat{g}_j(\widehat{\boldsymbol{z}}_{\text{new},[j]}) - g_j^*(\boldsymbol{z}_{\text{new},[j]}^*)\|^2)$$

$$= O(\alpha_{N,n}\xi_V^{-1}), \tag{C.4}$$

and

$$\mathbb{E}[V(y_i, \widetilde{y}_i(\boldsymbol{w})) - V(y_i, y_i^*(\boldsymbol{w}))] \leq \mathbb{E}^{1/2}(\|y_i\|^2) \max_{1\leq j\leq J} \mathbb{E}^{1/2}(\|\widetilde{g}_j(\widehat{\boldsymbol{z}}_{i,[j]}) - g_j^*(\boldsymbol{z}_{i,[j]}^*)\|^2) = O_p(\alpha_{N,n}) \tag{C.5}$$

Based on Assumption C.2, if $\mathbb{E}[V(y_i, \widetilde{y}_i(\boldsymbol{w})) - V(y_i, y_i^*(\boldsymbol{w}))] \leq \epsilon$, for any $\delta > 0$, we know that

$$\mathbb{P}\Big\{\xi_V^{-1} \sup_{\mathbf{w}\in\mathbb{W}} \Big| \frac{1}{n}\sum_{i=1}^{n}[V(y_i, \widetilde{y}_i(\boldsymbol{w})) - V(y_i, y_i^*(\boldsymbol{w}))] - \mathbb{E}[V(y_i, \widetilde{y}_i(\boldsymbol{w})) - V(y_i, y_i^*(\boldsymbol{w}))] \Big| > \delta\Big\}$$

$$\leq \xi_V^{-2}\delta^{-2}n^{-1}\text{Var}[V(y_i, \widetilde{y}_i(\boldsymbol{w})) - V(y_i, y_i^*(\boldsymbol{w}))]]$$

$$\leq \xi_V^{-2}\delta^{-2}n^{-1}c_2\epsilon^k. \tag{C.6}$$

Using equation C.5, equation C.6 and Assumption C.3, we know that

$$\sup_{\boldsymbol{w}\in\mathbb{W}} \frac{|CV(\boldsymbol{w}) - R_V^*(\boldsymbol{w})|}{R_V^*(\boldsymbol{w})}$$

$$\leq \xi_V^{-1}\sup_{\mathbf{w}\in\mathbb{W}} \Big| \frac{1}{n}\sum_{i=1}^{n}[V(y_i, \widetilde{y}_i(\boldsymbol{w})) - V(y_i, y_i^*(\boldsymbol{w}))] - \mathbb{E}[V(y_i, \widetilde{y}_i(\boldsymbol{w})) - V(y_i, y_i^*(\boldsymbol{w}))]\Big|$$

$$+ \xi_V^{-1}\sup_{\mathbf{w}\in\mathbb{W}} \Big| \frac{1}{n}\sum_{i=1}^{n}V(y_i, y_i^*(\boldsymbol{w})) - \mathbb{E}[V(y_i, y_i^*(\boldsymbol{w}))]\Big|$$

$$+ \xi_V^{-1}\sup_{\mathbf{w}\in\mathbb{W}} \Big| \mathbb{E}[V(y_i, \widetilde{y}_i(\boldsymbol{w})) - V(y_i, y_i^*(\boldsymbol{w}))]\Big|$$

$$= O_p(\xi_V^{-1}\alpha_{N,n}^{\kappa/2}\max\{n^{-1/2}, \alpha_{N,n}^{\kappa/2}\}) \tag{C.7}$$

Lemma 1 in (Zhang & Liu, 2023) states that if $\sup_{\boldsymbol{w}\in\mathbb{W}}|R_V(\boldsymbol{w}) - R_V^*(\boldsymbol{w})|/R_V^*(\boldsymbol{w}) = o(1)$ and $\sup_{\boldsymbol{w}\in\mathbb{W}}|CV(\boldsymbol{w}) - R_V^*(\boldsymbol{w})|/R_V^*(\boldsymbol{w}) = o_p(1)$ then the weight vector $\widehat{\boldsymbol{w}}$ selected by minimizing Eq. (3) satisfies $R_V(\widehat{\boldsymbol{w}})/\inf_{\boldsymbol{w}\in\mathbb{W}} R_V(\boldsymbol{w}) \to 1$ in probability. Therefore, we know when $\xi_V^{-1}\alpha_{N,n}^{\kappa/2}\max\{n^{-1/2}, \alpha_{N,n}^{1-\kappa/2}\} = o(1)$, combining equation C.4 and equation C.7, we have

$$\frac{R_V(\widehat{\boldsymbol{w}})}{\inf_{\mathbf{w}\in\mathbb{W}} R_V(\boldsymbol{w})} \to 1, \tag{C.8}$$

in probability.

From the proof in equation C.4, we know that

$$|\inf_{\mathbf{w}\in\mathbb{W}} R_V(\boldsymbol{w}) - \mathbb{E}(V(y_{\text{new}}, h^*(x_{\text{new}}, \boldsymbol{w}^*)))| \leq \sup_{\mathbf{w}\in\mathbb{W}}|R_V(\boldsymbol{w}) - R_V^*(\boldsymbol{w})| = O(\alpha_{N,n}). \tag{C.9}$$

Therefore, when $\xi_V^{-1}\alpha_{N,n} = o(1)$, equation C.9 goes with

$$\mathbb{E}(V(y_{\text{new}}, h^*(x_{\text{new}}, \boldsymbol{w}^*)))/\inf_{\mathbf{w}\in\mathbb{W}} R_V(\boldsymbol{w}) \to 1 \tag{C.10}$$

By equation C.8, equation C.10, and Assumption C.6, we know $\text{I}/\inf_{\mathbf{w}\in\mathbb{W}} R_L(\boldsymbol{w}) = o(1)$.

Similar to equation C.9, we know

$$|\mathbb{E}(V(y_{\text{new}}, \widehat{h}(x_{\text{new}}, \boldsymbol{w}^*))) - \mathbb{E}(V(y_{\text{new}}, h^*(x_{\text{new}}, \boldsymbol{w}^*)))| \leq \sup_{\mathbf{w}\in\mathbb{W}}|R_V(\boldsymbol{w}) - R_V^*(\boldsymbol{w})| = O(\alpha_{N,n}).$$
$$\tag{C.11}$$

Based on Assumption C.1 and equation C.9, we have

$$\inf_{\mathbf{w}\in\mathbb{W}} R_L(\boldsymbol{w}) - \inf_{\mathbf{w}\in\mathbb{W}} R_L^*(\boldsymbol{w}) = O_p(\alpha_{N,n}^{\mu}).$$

Since $\xi_L^{-1}\alpha_{N,n}^{\mu} = o(1)$ derived from Assumption C.6, we have $\inf_{\mathbf{w}\in\mathbb{W}} R_L^*(\boldsymbol{w})/\inf_{\mathbf{w}\in\mathbb{W}} R_L(\boldsymbol{w}) \to 1$. Therefore, we have

$$\frac{\text{II}}{\inf_{\mathbf{w}\in\mathbb{W}} R_L(\boldsymbol{w})} = \frac{\inf_{\mathbf{w}\in\mathbb{W}} R_L(\boldsymbol{w}) - \inf_{\mathbf{w}\in\mathbb{W}} R_L^*(\boldsymbol{w})}{\inf_{\mathbf{w}\in\mathbb{W}} R_L^*(\boldsymbol{w})} \times \frac{\inf_{\mathbf{w}\in\mathbb{W}} R_L^*(\boldsymbol{w})}{\inf_{\mathbf{w}\in\mathbb{W}} R_L(\boldsymbol{w})} = o(1).$$

Finally, we can obtain the Theorem 1.

$$\square$$

## C.2 ASSUMPTIONS AND PROOF OF THEOREM 2

### C.2.1 ASSUMPTIONS

Define $\mathbb{W}_s = \{\boldsymbol{w} \in \mathbb{W} : \sum_{j \notin \mathbb{D}} w_j = 1\}$ be a weight set that assigns all weights on the misspecified models .

**Assumption C.7.** *For every $\boldsymbol{w} \in \mathbb{W}$, there are $R_V^*(\boldsymbol{w}) \geq c_5 \|y_{\text{new}}^*(\boldsymbol{w}) - \mathbb{E}(y_{\text{new}}|x_{\text{new}})\|^2$, for some positive constant $c_5$.*

Assumption C.7 is similar to Assumption 5 in (Yu et al., 2025) which assumes the lower bound on the risk of true loss, and we focus on the surrogate one. Next, let us show how commonly used surrogate losses—squared error, logistic (cross-entropy), and hinge loss imply Assumption C.7. For squared-error loss where $V(y_{\text{new}}, y_{\text{new}}^*(\boldsymbol{w})) = L(y_{\text{new}}, y_{\text{new}}^*(\boldsymbol{w})) = (y_{\text{new}}^*(\boldsymbol{w}) - y_{\text{new}})^2$, By the usual orthogonality argument for squared error,

$$R_V^*(\boldsymbol{w}) = \mathbb{E}\big[(y_{\text{new}}^*(\boldsymbol{w}) - y_{\text{new}})^2\big] = \mathbb{E}\big[(y_{\text{new}}^*(\boldsymbol{w}) - \mathbb{E}[y_{\text{new}} \mid x_{\text{new}}])^2\big] + \mathbb{E}\big[(\mathbb{E}[y_{\text{new}} \mid x_{\text{new}}] - y_{\text{new}}^*(\boldsymbol{w}))^2\big].$$

Therefore, we have $R_V^*(\boldsymbol{w}) \geq \big\|y_{\text{new}}^*(\boldsymbol{w}) - \mathbb{E}[y_{\text{new}} \mid x_{\text{new}}]\big\|^2$.

For logistic loss with $y_{\text{new}} \in \{-1, 1\}$, we know $V(y_{\text{new}}, y_{\text{new}}^*(\boldsymbol{w})) = \log(1 + \exp(-y_{\text{new}}y_{\text{new}}^*(\boldsymbol{w})))$ and $R_V^*(\boldsymbol{w}) = \mathbb{E}[\log(1 + \exp\{-y_{\text{new}}y_{\text{new}}^*(\boldsymbol{w})\})]$. Since $\ell(m) = \log(1 + e^m)$ is strictly convex relative to $m$. Then

$$\log\big(1 + \exp(-y_{\text{new}}y_{\text{new}}^*(\boldsymbol{w}))\big) - \log\big(1 + \exp(-y_{\text{new}}\mathbb{E}(y_{\text{new}}|x_{\text{new}}))\big)$$
$$\geq c\big(y_{\text{new}}\mathbb{E}(y_{\text{new}}|x_{\text{new}}) - y_{\text{new}}y_{\text{new}}^*(\boldsymbol{w})\big)^2$$
$$\geq c\big(\mathbb{E}(y_{\text{new}}|x_{\text{new}}) - y_{\text{new}}^*(\boldsymbol{w})\big)^2,$$

for some constant $c > 0$. Then we have

$$R_V^*(\boldsymbol{w}) - \mathbb{E}\Big[\log\big(1 + e^{-y_{\text{new}}\mathbb{E}(y_{\text{new}}|x_{\text{new}})}\big)\Big] \geq c\mathbb{E}\Big[\big|y_{\text{new}}^*(\boldsymbol{w}) - \mathbb{E}(y_{\text{new}}|x_{\text{new}})\big|^2\Big].$$

Since the term $[\log\big(1 + e^{-y_{\text{new}}\mathbb{E}(y_{\text{new}}|x_{\text{new}})}\big)]$ is always larger than 0, then the logistic loss satisfies Assumption C.7.

For hinge loss with $y_{\text{new}} \in \{-1, +1\}$, we have $V\big(y_{\text{new}}, y_{\text{new}}^*(\boldsymbol{w})\big) = \max\{0, 1 - y_{\text{new}}y_{\text{new}}^*(\boldsymbol{w})\}$. By piecewise linearity and local strong-convexity that

$$R_V^*(\boldsymbol{w}) - \mathbb{E}[V(y_{\text{new}}, \mathbb{E}(y_{\text{new}}|x_{\text{new}})] \geq c\mathbb{E}[\big\|y_{\text{new}}^*(\boldsymbol{w}) - \mathbb{E}(y_{\text{new}}|x_{\text{new}})\big\|^2],$$

where the term $\mathbb{E}[V(y_{\text{new}}, \mathbb{E}(y_{\text{new}}|x_{\text{new}})]$ is a constant that does not depend on $\boldsymbol{w}$. Therefore, the hinge loss still satisfies Assumption C.7.

**Assumption C.8.** $\max\{\alpha_{N,n}^{\kappa/2}n^{-1/2}, \alpha_{N,n}^{\kappa}\}\{\inf_{\mathbf{w} \in \mathbb{W}_s} \|\sum_{j \notin \mathbb{D}} w_j[y_{i,[j]}^* - \mathbb{E}(y_{\text{new}}|x_{\text{new}})]\|^2\}^{-1} = o(1)$.

Assumption C.8 means that the divergence speed of minnimum risk of $L_2$ loss of averaging across all misspecified models is faster than $\max\{\alpha_{N,n}^{\kappa/2}n^{-1/2}, \alpha_{N,n}^{\kappa}\}$.

### C.2.2 PROOF OF THEOREM 2

*Proof.* Following arguments similar to those in equation C.7, we know that

$$CV(\boldsymbol{w}) - R_V^*(\boldsymbol{w}) = O_p(\max\{\alpha_{N,n}^{\kappa/2}n^{-1/2}, \alpha_{N,n}^{\kappa}\}). \tag{C.12}$$

Let

$$\widehat{\lambda}_j = \begin{cases} \widehat{w}_j/(1-\tau) & j \notin \mathbb{D} \\ 0 & j \in \mathbb{D}. \end{cases}$$

Based on Assumption C.7, we have

$$R_V^*(\widehat{\boldsymbol{w}}) \geq c_5\|y_{\text{new}}^*(\widehat{\boldsymbol{w}}) - \mathbb{E}(y_{\text{new}}|x_{\text{new}})\|^2 = (1 - \widehat{\tau})^2\|\sum_{j \notin \mathbb{D}} \widehat{\lambda}_j[y_{\text{new},[j]}^* - \mathbb{E}(y_{\text{new}}|x_{\text{new}})]\|^2.$$

$$\tag{C.13}$$

Define $\widetilde{\boldsymbol{w}}$ by a weight vector that $\sum_{j \in \mathbb{D}} \widetilde{w}_j = 1$. Then, we know $y_{\text{new}}^*(\widetilde{\boldsymbol{w}}) = \sum_{j \in \mathbb{D}} \widetilde{w}_j y_{[j]}^* = \sum_{j \in \mathbb{D}} y_{\text{new}} = y_{\text{new}}$. Therefore, $R^*(\widetilde{\boldsymbol{w}}) = 0$. Since $\widehat{\boldsymbol{w}}$ minimizes $CV(\boldsymbol{w})$, we get

$$0 = R_n^*(\widetilde{\boldsymbol{w}}) \geq C_n(\widehat{\boldsymbol{w}}) + O_p(\max\{\alpha_{N,n}^{\kappa/2} n^{-1/2}, \alpha_{N,n}^\kappa\})$$

$$\geq (1 - \widehat{\tau})^2 \inf_{\mathbf{w} \in \mathbb{W}_s} \| \sum_{j \notin \mathbb{D}} w_j [y_{\text{new},[j]}^* - \mathbb{E}(y_{\text{new}}|x_{\text{new}})]\|^2,$$

which implies that $\widehat{\tau} \to 1$ and completes the proof. $\qquad\square$

## APPENDIX D FOUNDATION REPRESENTATION SETS AND CANDIDATE REPRESENTATION SETS

In this section, we clarify the conceptual difference between the $M$ foundation representation sets and the $J$ candidate representation sets, and explain when the theoretical analysis can be formulated directly in terms of the $J$ candidate sets.

### D.1 CONCEPTUAL DIFFERENCE

Foundation representation sets are the primitive objects obtained from foundation representation learning (FRL) models trained on unlabeled data, or from pretrained models. They provide the structural building blocks from which candidate representation sets are derived.

To better illustrate the distinction, consider a toy example with $M = 3$ FRL models that produce $M$ representation sets $\{z_{(1)}, z_{(2)}, z_{(3)}\}$. Suppose the ground-truth target depends only on the first two sets:

$$y = g^\star(z_{(1)}, z_{(2)}) + \varepsilon,$$

while $z_{(3)}$ is essentially noise. If we only fit one prediction model per FRL, i.e.

$$\hat{g}_1(z_{(1)}), \quad \hat{g}_2(z_{(2)}), \quad \hat{g}_3(z_{(3)}),$$

then $\hat{g}_1$ and $\hat{g}_2$ are misspecified (each sees only part of the relevant information, e.g., using only $z_{(1)}$ to predict $y$), and $\hat{g}_3$ is almost uninformative.

In our framework, we derive $J$ candidate representation sets from these $M$ FRLs. For example, using the Foundation Set Combinations construction in Example 1 of the paper, we obtain the $J = 2^M - 1$ candidates

$$(1), (2), (3), (1,2), (1,3), (2,3), (1,2,3),$$

where "$(1,2)$" denotes the concatenation of $z_{(1)}$ and $z_{(2)}$, etc. In this example, candidate construction creates *correctly specified* models such as $(1,2)$ and $(1,2,3)$ out of a starting point where each single-FRL model is misspecified or uninformative. Among these candidates, $(1,2)$ contains exactly the sufficient information for $y$, while the full fusion $(1,2,3)$ adds redundant or noisy dimensions and is more prone to overfitting. PEARL operates on this candidate family by learning weights over the $J$ candidate models from cross-validated predictions.

### D.2 IMPLICATIONS FOR THEORETICAL ANALYSIS

If one treats the FRLs as fixed and focuses solely on the averaging stage (the downstream task-learning step), then the framework can indeed be formulated directly in terms of the $J$ candidate representation sets. Concretely, one can assume a collection of fixed FRL models $\{f_m\}_{m=1}^M$, define

$$z_{i,(m)} = f_m(x_i),$$

and then work only with the derived candidate representations $z_{i,[j]}$ for $1 \leq j \leq J$. This corresponds to a practical scenario where one uses $M$ pretrained models and regards them as fixed foundation encoders.

However, the goal of the theory in this paper is to analyze the full pipeline of the proposed method, starting from representation learning, where each FRL $f_m$ can itself be trained from unlabeled data $D^u$. In this more general setting, the learned FRLs $\hat{f}_m$ and the induced representations

$$\hat{z}_{i,(m)} = \hat{f}_m(x_i)$$

are random objects, and their estimation error and randomness must be accounted for in the asymptotic analysis. Our theoretical framework is therefore designed to cover both (i) fixed pretrained FRLs $f_m$ and (ii) learned FRLs $\hat{f}_m$, so that it applies to settings where representation learning is part of the procedure as well as those where one starts from pretrained foundation encoders and works directly with the $J$ candidate representation sets.

# APPENDIX E   IMPLEMENTATION DETAILS OF EXPERIMENTS

This section shows the implementation details of our experiments. Each experiment's setup is tailored to the characteristics of the dataset and the objectives of the study.

## E.1   SUMMARY OF EXPERIMENTS

Table E.5 outlines the experiments and the associated FRL methods; implementation details are provided in the following sections.

Table E.5: Summary of FRL methods applied to different tasks and datasets in our experiments.

| Task Type & Dataset | FRL Methods |
| --- | --- |
| Simulated data in equation 7: A synthetic regression task with a nonlinear data generating process. | KPCA (Linear Kernel) 
 KPCA (Gaussian Kernel) 
 KPCA (Polynomial Kernel) 
 KPCA (Sigmoid Kernel) 
 KPCA (Cosine Kernel) |
| CIFAR-10 (Krizhevsky, 2009): An image classification task that classifies color images into one of ten categories. | AE 
 VAE 
 SimCLR (CNN encoder) (Chen et al., 2020) 
 SimCLR (Resnet encoder) |
| CIFAR-10: pretrained FRLs. | torchvision[E.1] (resnet18.tv_in1k) (Paszke et al., 2019) 
 torchvision (torchvision:resnet50) 
 timm (resnext50_32x4d.ra_in1k) (Wightman, 2019) 
 timm (convnext_tiny.fb_in1k) (Liu et al., 2022) |
| CIFAR-100: pretrained FRLs. | timm (vit_base_patch14_dinov2.lvd142m) (Oquab et al., 2023) 
 timm (vit_base_patch16_224.augreg_in1k) (Dosovitskiy et al., 2021) 
 timm (swin_base_patch4_window7_224.ms_in22k_ft_in1k) (Liu et al., 2021) 
 timm (vit_base_patch16_224.dino) (Caron et al., 2021) |
| Polarity (Pang & Lee, 2004): A text classification task that predicts sentiment polarity (positive or negative) from movie reviews. | CountVectorizer 
 TF-IDF Vectorizer 
 Word2Vec (Mikolov et al., 2013) 
 Doc2Vec (Le & Mikolov, 2014) 
 GloVe (Pennington et al., 2014) 
 BERT (Devlin et al., 2019) |
| Polarity: LLM-based FRLS | Qwen/Qwen2.5-1.5B-Instruct (Qwen Team, 2025) 
 meta-llama/Llama-3.2-1B-Instruct (AI@Meta Llama Team, 2024) 
 openai-community/gpt2-xl (Radford et al., 2019) |
| Wine Review: A multi-modal regression task that predicts the quality of wines. | PCA (Numerical Modality) 
 VAE (Numerical Modality) (Kingma & Welling, 2014) 
 BERT (Textual Modality) 
 GPT-2 (Textual Modality) (Radford et al., 2019) |

## E.2   EXPERIMENTS ON CIFAR-10 AND CIFAR-100 DATASETS

**Dataset Preparation and Normalization**   The experiment uses the CIFAR-10 dataset with the standard per-channel normalization used in prior work. Images are first converted to tensors (values in $[0, 1]$) and then standardized with the dataset means $(0.4914, 0.4822, 0.4465)$ and standard deviations $(0.2470, 0.2435, 0.2616)$.

**Representation Learning with Autoencoder** For the Autoencoder, we use a lightweight, custom ResNet-style encoder. It has a $3\times3$ convolutional stem, followed by three residual stages with channel widths $(32, 64, 128)$, where each stage contains two residual units, and spatial downsampling is applied at the start of the last two stages. A global average pooling and a small MLP head map features to a 32-dimensional representation. The decoder is a convolutional (transpose-conv) network that upsamples back to $32 \times 32 \times 3$, mirroring the encoder in capacity. We train with Mean Squared Error (MSE) between reconstructions and the normalized inputs, i.e., the loss is computed in the standardized pixel space. Optimization uses Adam with learning rate $10^{-3}$, batch size 256, for 50 epochs with early stopping (patience $= 3$).

**Representation Learning with VAE** The VAE shares the same ResNet encoder trunk as AE but produces $\mu$ and $\log \sigma^2$ for a 32-dimensional Gaussian latent. A sample $z$ is drawn via the reparameterization trick and passed through the same CNN decoder used by the Autoencoder. The objective is $\mathcal{L} = \mathrm{MSE}(x, \hat{x}) + \beta \, \mathrm{KL}\big(q(z|x) \,\|\, \mathcal{N}(0, I)\big)$ with $\beta = 1$. We also train with Adam ($10^{-3}$), batch size 256, for 50 epochs with early stopping (patience $= 3$).

**Representation Learning with SimCLR** We consider two SimCLR encoders: a simple CNN (two stride-2 conv blocks followed by FC layers) and a ResNet trunk same as AE. Both are followed by a 2-layer MLP projection head with BatchNorm. Training follows the vanilla objective (Chen et al., 2020) with temperature $\tau = 0.5$. The data augmentation pipeline for each view includes: RandomResizedCrop($32$, scale $= (0.2, 1.0)$), random horizontal flip, ColorJitter $(0.8, 0.8, 0.8, 0.2)$ applied with probability $0.8$, random grayscale with probability $0.2$, Gaussian blur (kernel size 5, $\sigma \in [0.1, 2.0]$) applied with probability $0.5$, then the same per-channel normalization as above. We optimize with Adam, base learning rate $3 \times 10^{-3}$ and a $10\%$ warm-up followed by cosine decay, batch size 512, for 20 epochs.

**CIFAR-10 with Pretrained ResNet/ConvNeXt Backbones** For the additional CIFAR-10 experiments with stronger FRLs, we instantiate $M = 4$ pretrained convolutional backbones as FRLs. Specifically, we use `resnet18.tv_in1k` and `torchvision:resnet50` from `torchvision`, and `timm:resnext50_32x4d.ra_in1k` and `timm:convnext_tiny.fb_in1k` from `timm`. We follow the default preprocessing pipeline of each backbone (including resizing to the input resolution and ImageNet-style normalization) and extract a single feature vector per image by applying global average pooling to the final convolutional feature map (for ResNet/ResNeXt/ConvNeXt). These feature vectors are treated as FRL representations and used to construct candidate models exactly as in the base CIFAR-10 configuration. The backbones are kept frozen and we reuse the same downstream classifiers and training protocol as in the main CIFAR-10 experiments for PEARL and all baselines.

**CIFAR-100 with Pretrained ViT/Swin Backbones** For the CIFAR-100 experiments, we use $M = 4$ pretrained transformer-based models from `timm` as FRLs: `vit_base_patch14_dinov2.lvd142m`, `vit_base_patch16_224.augreg_in1k`, `swin_base_patch4_window7_224.ms_in22k_ft_in1k`, and `vit_base_patch16_224.dino`. CIFAR-100 images are resized to the input resolution required by each backbone and preprocessed using the corresponding default normalization. For ViT-based models we use the `[CLS]` token embedding as the FRL representation, and for Swin we apply global average pooling over the final stage features. These representations serve as the inputs to candidate models constructed analogously to the CIFAR-10 case. All backbone parameters remain frozen; only the downstream classifiers used by PEARL and the competing methods are trained, with the same optimization hyperparameters as in the main CIFAR-100 experiments.

### E.3 EXPERIMENTS ON POLARITY DATASET

**Dataset Preparation and Tokenizer** The Polarity dataset, comprised of text documents with positive or negative sentiment labels, is preprocessed to remove special characters and stopwords, preparing it for representation learning. A standard NLP tokenizer is employed for most techniques, while BERT benefits from its specialized tokenizer to best utilize its pretraining. This preprocessing step ensures the data is in a suitable format for subsequent representation learning methods.

**Representation Learning with Vectorization Techniques** We treat some vectorization techniques as representation learning methods to transform textual data into numerical vectors. Specifically, we employ the CountVectorizer and TF-IDF Vectorizer from the scikit-learn library, configuring both to limit the vocabulary to the top 10,000 most frequent terms in the dataset. The CountVectorizer simply tallies word frequencies, while the TF-IDF Vectorizer additionally considers the inverse document frequency of terms, offering a measure of how informative each word is across the corpus. Therefore, there is no extra training that needs configuration.

**Representation Learning with other Methods** We consider other methods including Word2Vec, Doc2Vec, GloVe, and BERT as representation learning methods for text documents. Doc2Vec and Word2Vec are implemented via the Gensim library to process an entire document to yield 100-dimensional embeddings, where Word2Vec's document vectors result from averaging token-level embeddings. The GloVe embeddings are sourced from the official GloVe website[E.2], specifically the glove.6B.300d variant, offering pre-trained 300-dimensional vectors. BERT embeddings are obtained from the `bert-base-uncased` model available through Hugging Face, extracting document representations by averaging the outputs from the final hidden layer. The configurations of each model not explicitly mentioned follow the default settings of their corresponding libraries.

**Representation Learning with Large Language Models** For the LLM-based FRLs used in the Polarity experiments, we rely on Hugging Face's `AutoTokenizer` and `AutoModel` to load three pretrained models: `Qwen/Qwen2.5-1.5B-Instruct`, `meta-llama/Llama-3.2-1B-Instruct`, and `openai-community/gpt2-xl`. For each review, we first tokenize the raw text without truncation and split the resulting token sequence into contiguous chunks whose lengths are bounded by a maximum sequence length (512 tokens). We use the last hidden layer to obtain a representation for each chunk. For encoder-style architectures, we apply attention-mask–weighted mean pooling over all token positions; for decoder-only architectures, we use the hidden state of the last non-padding token. If a review produces multiple chunks, their chunk-level embeddings are averaged to obtain a single fixed-dimensional document representation.

E.4    EXPERIMENTS ON WINE REVIEW DATASET

**Data Preprocessing** We begin by processing the structured features, which include `price`, `variety`, `country`, `province`, `region_1`, and `region_2`. Missing values in the `price` field are imputed using the median, while missing categorical entries are filled with a default value ("unknown"). Categorical features are transformed via one-hot encoding, and continuous features are normalized to zero mean and unit variance. Representation learning is performed on an unlabeled subset of the dataset, denoted as $D^u$, containing $n_{\mathrm{un}} = 10000$ samples randomly chosen from the first 100000 records. The prediction task involves learning a linear regression model to predict the wine quality score using the candidate representation sets. The labeled set $D^l$ is also randomly drawn from the first 100000 records of the dataset, with $n$ varying across $\{100, 200, 500, 1000, 2000\}$. The test set consists of the last 5000 records of the dataset.

**Representation Learning for Numerical Modality** For the numerical modality, we apply two complementary representation learning methods. First, Principal Component Analysis (PCA) is employed to reduce the structured feature space to 50 dimensions. Second, a Variational Autoencoder (VAE) (Kingma & Welling, 2014) is used to learn a compact latent representation. The VAE is configured with a latent dimension of 32 and an encoder consisting of three linear layers with hidden sizes of 128, 128, and $2 \times 32$ (for the mean and log-variance). It is trained using the Adam optimizer with a learning rate of 0.001 over 20 epochs, with early stopping triggered if the validation loss does not improve for 5 consecutive epochs.

**Representation Learning for Textual Modality** For the textual modality, we derive representations using two pretrained language models from Hugging Face. Both models obtain embeddings by averaging the outputs of their last hidden states. For BERT, we use the `bert-base-uncased` model, with inputs tokenized to a maximum sequence length of 512. Similarly, GPT-2 (Radford et al., 2019) is used with tokenization limited to a maximum of 512 tokens.

---

[E.2] https://nlp.stanford.edu/projects/glove/

### E.5 EXPERIMENTS ON MNIST DATASET

We follow the same training protocol as on CIFAR-10 with the some changes specific to MNIST. Inputs are $28\times28$ grayscale images, converted to tensors in $[0, 1]$ and standardized using the dataset statistics (mean $0.1307$, std $0.3081$). We train four unsupervised encoders: an Autoencoder with a CNN backbone (AE–CNN), SimCLR with a CNN backbone (SimCLR–CNN), SimCLR with an MLP backbone (SimCLR–MLP), and a Variational Autoencoder with a CNN backbone (VAE–CNN). For AE/VAE we use the same MSE reconstruction objective as those used in CIFAR-10 experiments. The data augmentation pipeline of SimCLR is RandomResizedCrop($28$, scale $= (0.6, 1.0)$, RandomAffine (degrees $30$, translate $(0.2, 0.2)$, scale $(0.7, 1.3)$, shear $10$), RandomPerspective ($p = 0.2$), Gaussian blur (kernel $3$, $\sigma \in [0.1, 1.2]$), then the same normalization). Unless otherwise noted, we adopt the same optimization hyperparameters as in the CIFAR-10 experiments.