# OpenReview forum: "Performance-Enhanced Aggregated Representation Learning"
_ICLR.cc/2026/Conference — Submitted to ICLR 2026_

### Official Review · Reviewer_u8HM · 2025-10-25

**Soundness:** 3
**Presentation:** 4
**Contribution:** 3
**Rating:** 6
**Confidence:** 3

**Summary:**

This paper focuses on the aggregation of representations of raw input data. They propose a method to combine the different represantations and the possible sets of representations that can be extracted from the original representations. Normally, one can start with finding the optimal weights by minimizing the loss function between the actual labels and the predicted labels, where the predicted labels are coming from the models that are trained on the entrire labeled dataset on top the specific representations. However, instead of doing this the authors suggest that when we use predicted labels where the prediction comes from K-fold cross validation (basically the model that makes the prediction on a sample should not be trained on this sample), the overall performance improves. It is also suggested that instead of the actual loss function using a surrogate loss function can enhance the smoothness and and induce convexity.

**Strengths:**

This work focuses on very generic setup of aggregation of different representation learnings and it shows that PEARL can be used for variety of machine learning projects that involve LLMs as well as classical machine learning methods such as linear models, decisiion trees etc. as long as the representations of the raw data can be found.

Even though very similar methods are being used in machine learning competitions, the authors are suggesting that they prove PEARL reaches optimal performance on downstream tasks under certain conditions.

Authors also compare their results with different aggregation methods on several datasets as well as a synthetic dataset, so that they also provide experimental results in addition to theoretical analysis.

**Weaknesses:**

Even though having analysis on specifically multiple representations can be new, very similar methods which involves k-fold cross validation based weighting in the ensembling phases are commonly used in machine learning competitions. As far as I understand, the very high level idea of this paper is this : "When it comes to the combining features or ensembling of different predictions, it is better to find weights based on K-fold based predictions -not training a model and use its own predictions of the same saples-". So the core idea is not super-new.

**Questions:**

1 - I wanna learn the difference between the M representations from FRL models and J candidate representation sets. So, I do understand that we have M model representations and we can extract J different candidate representation sets from these M representations. However, theoretically, do we really need to mention that we have M representations initially ? We can basically think of we have J different representations where these representations can have similarities/common-subvectors without even mentioning the initial M models right ? I wanna make sure that I do not miss if this method exploits the information that these J candidate sets are constructed from M model representations.

---

> ### Author Response · Authors · 2025-11-21
>
> We sincerely thank you for your constructive comments. We truly appreciate the time and effort you’ve dedicated to reviewing our submission. We hope that our clarifications address your concerns, and we would be happy to provide further discussion or additional details if needed.
>
> > **Weakness 1**: *Even though having analysis on specifically multiple representations can be new, very similar methods which involves k-fold cross validation based weighting in the ensembling phases are commonly used in machine learning competitions. As far as I understand, the very high level idea of this paper is this : "When it comes to the combining features or ensembling of different predictions, it is better to find weights based on K-fold based predictions -not training a model and use its own predictions of the same saples-". So the core idea is not super-new.*
>
>
> **Response 1**：Thank you for your comments. We agree that using $K$-fold cross-validation (CV) is a well-known idea, and we explicitly build on this principle. However, our contribution is not merely to restate this idea. We clarify the differences below.
>
> - **Methodological novelty**. As the reviewer notes, our methodology focus representation aspects. Specifically, PEARL averages task-specific predictors that are built on top of heterogeneous deep representations, rather than directly averaging arbitrary end-to-end predictors. Therefore, the learned weights quantify how much each representation contributes to the downstream task.
>
> - **Theoretical novelty**. Although $K$-fold based ensembling for complex models is numerically effective in practice, there is no sufficient theoretical understanding of why such weighting should work for deep models, nor how $K$-fold surrogate losses relate to the true downstream loss. Our work fills this gap by establishing asymptotic optimality with respect to the original downstream loss, and consistency of the learned weights.

---

> > ### Author Response · Authors · 2025-11-21
> >
> > > **Q1**: *I wanna learn the difference between the M representations from FRL models and J candidate representation sets. So, I do understand that we have M model representations and we can extract J different candidate representation sets from these M representations. However, theoretically, do we really need to mention that we have M representations initially? We can basically think of we have J different representations where these representations can have similarities/common-subvectors without even mentioning the initial M models right? I wanna make sure that I do not miss if this method exploits the information that these J candidate sets are constructed from M model representations.*
> >
> > **Response to Q1**: Thank you for the insightful question. We would like to first clarify the difference between $M$ foundatation representations and $J$ candidate representations, and then state when the theoretical analysis can start right from $J$ candidate representations.
> >
> > - Conceptual clarity. Foundation representation sets are the primitive objects obtained from FRL models trained on unlabeled data, or from pretrained models. They provide the structure and source from which candidate sets are derived.
> >
> >   We provide a toy example to better illustrate the difference. Suppose we have $M = 3$ FRL models that produce $M$ representation sets $\{z_{(1)}, z_{(2)}, z_{(3)}\}$, and the ground truth target depends only on the first two: $$ y = g^\star\big(z_{(1)}, z_{(2)}\big) + \varepsilon,$$ while $z_{(3)}$ is essentially noise. If we only fit one prediction model per FRL, i.e., $$\hat g_1(z_{(1)}),\quad \hat g_2(z_{(2)}),\quad \hat g_3(z_{(3)}),$$ then $\hat g_1$ and $\hat g_2$ are misspecified (each sees only part of the relevant information, e.g., training a model that uses only $z_{(1)}$ to predict $y$), and $\hat g_3$ is almost useless.
> >
> >   In our framework, we can derive $J$ candidate representation sets from these $M$ FRLs. For example, if we use the Foundation Set Combinations method in Example 1 of the paper, we construct candidates $$ (1),\ (2),\ (3),\ (1,2),\ (1,3),\ (2,3),\ (1,2,3),$$
> >   where “$(1,2)$” denotes the concatenation of $z_{(1)}$ and $z_{(2)}$, etc. **The candidate construction creates correctly specified models, such as $(1,2)$ and $(1,2,3)$, out of a starting point where each single-FRL model is misspecified or uninformative.** Among these candidates, $(1,2)$ contains exactly the sufficient information for $y$, while the full fusion $(1,2,3)$ adds many redundant or noisy dimensions and is more prone to overfitting. PEARL operates on this candidate family: it learns weights over the $J$ candidates from cross-validated predictions.
> >
> >
> > - Theoretical analysis. If one treats the FRLs as fixed and focuses on the averaging stage (downstream task learning process), we agree that the framework can indeed be formulated directly in terms of $J$ candidate representation sets. Specifically, one can assume a collection of fixed FRL models $\{f_m\}$, define $ z_{i,(m)} = f_m(x_i)$, and then work only with the derived candidates $z_{i,[j]}$ for $1\le m \le M$ and $1\le j \le J$. This can correspond to the practical situation where one uses $M$ pretrained models and regards them as fixed FRL models.
> >
> >   However, the goal of the theory in this paper is to   analyze the full pipeline of the proposed method starting from representation learning, where each FRL $f_m$ can itself be trained from unlabeled data $D^u$. In this more general setting, the learned FRL models $\hat f_m$ and the induced representations $\hat{z}_{i,(m)}=\hat{f}_m (x_i)$ are random objects, and their estimation error and randomness must be accounted for in our asymptotic analysis. Our theoretical framework is designed to cover both (i) the fixed pretrained FRLs $f_m$ and (ii) the learned FRLs $\hat{f}_m$, so that it applies to settings with pretrained encoders as well as those where representation learning is part of the overall procedure.

---

### Official Review · Reviewer_ioJk · 2025-10-29

**Soundness:** 2
**Presentation:** 2
**Contribution:** 1
**Rating:** 2
**Confidence:** 4

**Summary:**

This work proposes the following ensembling method:
1. Train a bunch of encoders using different techniques
2. Combine the dimensions of these trained embeddings. Each combination is a candidate
3. Train a predictor for each candidate
4. Obtain linear weights for the candidates using e.g. cross validation

The authors test the method on synthetic datasets, CIFAR-10, and some tabular datasets

**Strengths:**

1. The writing is fine. It is not particularly hard to understand the paper
2. Learning the linear weights using cross validation seems reasonable

**Weaknesses:**

This manuscript does not have enough novelty and sophistication to reach bar of ICLR. While it might be sufficient for a course project, I don't think it should be accepted by a top venue like ICLR.

The proposed method is essentially linearly ensembling a bunch of models, where the linear weights are obtained via cross valiation. All of these are pretty standard. I feel the only thing that might be novel is the way the method chooses the candidates, which is by combining the dimensions from different encoders. However, there are two problems:
1. From Section 2.2, it is not clear to me how the candidates should be chosen. Obviously one cannot choose every possible combination, for that will give us an exponential number of candidates. But if we use a subset, it is not clear how this subset should be chosen. Note that for a deep embedding, multiplying it by an invertible matrix typically does not change the representation. This means that if we choose one subset, then by multiplying a permutation matrix, we end up in another subset. It is also not clear to me how the "domain knowledge" can be used in choosing this subset.
2. It is not clear why we should choose a subset of dimensions at all. Equivalently, one can use all dimensions of the embeddings, and multiply it by a masking matrix that masks certain dimensions to zero. In this case, why not learn this masking matrix, when learning the downstream predictor? What does this seemingly arbitrary way of masking certain dimensions buy us?

The theoretical analysis presented is very elementary, and the experiment section does not show strong evidence that doing what the authors suggest would help in real use cases. The experimental results are very weak. For example, in Table 2, why should any one care if the accuracy on CIFAR-10 is only 62%?

In conclusion, I recommend rejecting this submission.

**Questions:**

N/A

---

> ### Author Response · Authors · 2025-11-21
>
> We sincerely thank you for your detailed comments. We truly appreciate the time and effort you have dedicated to reviewing our submission. We hope that our clarifications address your concerns, and we would be happy to provide further discussion or additional details if needed.
>
> We would like to restate the key novelty of our work. Our main contribution is a general model-averaging framework with theoretical guarantees for heterogeneous deep learning models under general (possibly non-convex) losses, together with a theoretical analysis of the surrogate-to-original loss gap in the resulting model-averaging properties. Existing theoretically grounded model-averaging methods mainly focus on traditional statistical models and convex, computationally tractable losses.
>
>
>
>
>
>
> > *1. From Section 2.2, it is not clear to me how the candidates should be chosen. Obviously one cannot choose every possible combination, for that will give us an exponential number of candidates. But if we use a subset, it is not clear how this subset should be chosen.*
>
> **Response**：Thank you for your comment about the candidate construction process. We clarify that exponential combinations are not required in practice, such as $J=15$ in CIFAR-10 dataset in our empirical study. Example 1 in the paper gives three independent strategies. Importantly, the framework is not restricted to the specific constructions listed in Example 1: the masking-based constructions suggested by you are in fact fully compatible with our setup and can be included as candidates and treated in exactly the same way by PEARL (see the response to “Masking” question below for a detailed discussion).
>
>
>
>
> > *Note that for a deep embedding, multiplying it by an invertible matrix typically does not change the representation. This means that if we choose one subset, then by multiplying a permutation matrix, we end up in another subset.*
>
> **Response**: Indeed, different permutations or rotations of the same latent deep representation can be viewed as equivalent. Our framework does not attempt to enumerate all such reparameterizations. Instead, we fix each FRL model's output in the concrete coordinate system produced by training and construct candidates within this coordinate system. For example, if $x = (x_1,x_2,x_3)$ and a trained FRL model $\hat f_1$ produces $\hat z_{(1)} = \hat f_1(x) = (x_1, x_2, x_1^2)$, while another FRL $\hat f_2$ produces $\hat z_{(2)} = \hat f_2(x) = (x_2,  x_3^2)$, then the concatenated representation is $\hat z_{full} = (x_1, x_2,  x_1^2,  x_2,  x_3^2)$. In our analysis we simply regard this concrete ordering of coordinates in $\hat z_{(1)}$ and $\hat z_{(2)}$ (and also in $\hat z_{{full}}$) as the realized coordinate system produced by training, and we do not treat permutations of these coordinates as distinct candidates. The theoretical results depend only on the predictive risks of the candidate models, not on how their internal coordinates are parameterized. If two candidates differ only by a permutation of coordinates, there are regarded as the same candidate and their cross-validated losses and predictive risks will be essentially the same.
>
> > *It is also not clear to me how the "domain knowledge" can be used in choosing this subset.*
>
> **Response**: In deep representation settings, domain knowledge can be encoded by designating some FRLs as “always-used” and others as “optional”. For example, in practice one may believe that a particular FRL model is highly reliable (e.g., a strong backbone in vision). In our framework, one can always include the representation from this trusted FRL in every candidate representation, while only selectively adding other FRL models in some candidates. This is analogous to the asset-pricing example we mentioned (where factors such as {MKT, SMB, HML} are always retained and only machine-learning latent factors are varied), but it applies equally to deep representation settings.

---

> ### Author Response · Authors · 2025-11-21
>
> > It is not clear why we should choose a subset of dimensions at all. Equivalently, one can use all dimensions of the embeddings, and multiply it by a masking matrix that masks certain dimensions to zero. In this case, why not learn this masking matrix, when learning the downstream predictor? What does this seemingly arbitrary way of masking certain dimensions buy us?
>
> **Response**： We thank the reviewer for raising this point about masking-based alternatives. We think the masking-matrix idea with all dimensions of the embeddings can be interpreted in our notation as follows. Let $z_{{full}}$ denote the concatenated representation of all $M$ FRL models,
>     $$
>     z_{full} = (z_{(1)},\dots,z_{(M)}),
>     $$
>     and consider a masking (or more generally, linear) matrix $W$ so that $W z_{{full}}$ zeros out or re-weights certain coordinates. Learning $(W,g)$ jointly on the labeled data still corresponds to fitting one single predictor on $z_{{full}}$: the matrix $W$ can be absorbed into the first layer of $g$, so this is just a richer parametrization of a single downstream model on the fused embedding. This is exactly the same paradigm as our “fusion” baseline, where we concatenate representations of all FRL models and train one downstream model on the full representation (in our implementation, when we use MLP as $g$, then the first linear layer of $g$ plays the role of such a $W$). Empirically, PEARL outperforms this “use all dimensions and learn a single predictor” strategy systematically in our experiments.
>
>   Conceptually, PEARL can address model uncertainty, not only feature selection within one fixed representation.
>
>   - A specific mask $W$ applied to $z_{{full}}$ is equivalent to choosing one representation candidate $z_{[j]} = W z_{{full}}$ and then fitting a predictor on it, i.e., a form of model selection. In contrast, PEARL treats multiple such candidates (including the full fusion, partial concatenations, or low-dimensional subspaces) as separate models and performs model averaging across them. The model-averaging literature shows that averaging over several plausible models is typically more robust than committing to a single large or single sparsified model, especially under representation misspecification and limited labeled data.
>   - From the perspective of our framework, learnable masked methods can be naturally incorporated as additional candidates rather than as a single learned mask. In particular, our formulation does not require that different candidates use distinct representations, nor that the downstream predictors $g_j$ share the same architecture. One can, for example, treat several models that all take the full concatenated embedding $z_{{full}}$ as input but differ in their internal masking mechanisms or hyperparameters as distinct candidates: there exists a subset of indices $\mathcal J_{{mask}} \subseteq \{1,\dots,J\}$ such that for all $j \in \mathcal J_{{mask}}$ we set $z_{[j]} = z_{{full}}$, while the associated predictors $g_j$ implement different masking schemes or architectures. PEARL is then applied exactly as before to learn weights over this enlarged family of candidates. In this sense, learnable mask-based predictors fit directly into our representation–averaging framework, without any change to the algorithmic or theoretical development.
>
> > *The theoretical analysis presented is very elementary.*
>
> **Response**: Thank you for your comment on our theoretical results.  First, we would like to state the significance of our asymptotic optimality theory. These results demonstrate that our method is asymptotically superior to simple fusion schemes, equal-weight combinations, and any individual candidate model. Moreover, selecting a single candidate model is equivalent to using a one-hot weight vector. Since our asymptotic optimality holds over the full weight space, it automatically implies superiority over all one-hot selections as a special case.
>
> Second, We would like to clarify the main technical difficulty in our theory. Our goal is to establish asymptotic optimality with respect to the original task loss, while in practice the weights are estimated by minimizing a surrogate loss. This induces a nontrivial gap between the surrogate-loss–based weight estimation procedure and the original-loss–based risk criterion.  More concretely, our analysis must simultaneously control
>
> (1) The discrepancy between the weights that are optimal for the surrogate loss and those that would be optimal for the original loss.
>
> (2) The gap between the asymptotic optimality established under the surrogate risk and the desired asymptotic optimality under the original risk.
>
>   To the best of our knowledge, existing model-averaging results either work directly with the original (often quadratic) loss or analyze surrogate risks without having to bridge this two-level gap.

---

> ### Author Response · Authors · 2025-11-21
>
> > *the experiment section does not show strong evidence that doing what the authors suggest would help in real use cases. The experimental results are very weak. For example, in Table 2, why should any one care if the accuracy on CIFAR-10 is only 62%?*
> >
>
> **Response**: Thank you for your concerns about our experiments.  We agree that some CIFAR-10 accuracies in the original version look low compared with SOTA results. This is mainly because our primary goal is to evaluate the effectiveness of representation aggregation, not to push SOTA on a specific benchmark. In addition, in the CIFAR-10 experiments we deliberately adopt a setting close to semi-supervised learning: we first train the FRL models in an unsupervised/self-supervised manner, and then train the classifier using only a small labeled subset (0.6%–6% of the entire CIFAR-10 dataset). This setup reflects many real-world scenarios with abundant unlabeled data but limited labels.
>
>   To address this concern, we have added new experiments in which we replace the current self-trained FRL models with stronger pretrained backbones (ResNets and ViTs from torchvision/timm) on CIFAR-10 and CIFAR-100, still under the same limited-label regime. The following table reports the results on CIFAR-10 with pretrained ResNet-based FRL models (we set $M=4$ and use `resnet18.tv_in1k`, `torchvision:resnet50`, `timm:resnext50_32x4d.ra_in1k`, and `timm:convnext_tiny.fb_in1k`):
>
> | CIFAR-10     | n    | Fusion          | Best            | SA-FRL         | SA-cand         | MS             | PEARL              |
> |-------|------|-----------------|-----------------|-----------------|-----------------|-------------------|--------------------|
> | Acc   | 1500 | 88.953 (0.301)  | 90.371 (0.271)  | 90.431 (0.277)  | 89.789 (0.281)  | *90.463 (0.280)*  | **91.340** (0.231) |
> |       | 3000 | 90.345 (0.228)  | 91.095 (0.220)  | 91.557 (0.200)  | 91.094 (0.202)  | *91.583 (0.250)*  | **92.316 (0.217)** |
> | CE    | 1500 | 0.362 (0.011)   | 0.294 (0.007)   | 0.307 (0.005)   | 0.295 (0.006)   | *0.289 (0.008)*   | **0.259 (0.006)**  |
> |       | 3000 | 0.333 (0.009)   | 0.280 (0.006)   | 0.275 (0.004)   | *0.259 (0.004)* | 0.262 (0.007)     | **0.231 (0.004)**  |
>
>   If we use stronger model like ViTs ($M=4$, we use models from timm: `vit_base_patch14_dinov2.lvd142m`, `vit_base_patch16_224.augreg_in1k`, `swin_base_patch4_window7_224.ms_in22k_ft_in1k`,`timm:vit_base_patch16_224.dino`), we can obtain the following results for the more complex dataset CIFAR-100:
> | CIFAR-100 | n    | Fusion        | Best          | SA-FRL        | SA-cand        | MS          | PEARL          |
> |--------|------|---------------|---------------|----------------|----------------|----------------|----------------|
> | Acc    | 1500 | 84.013 (0.547) | 84.936 (0.394) | 83.572 (0.534) | 84.364 (0.489) | *85.688* (0.434) | **86.241** (0.411) |
> |     | 3000 | 87.178 (0.303) | 87.234 (0.283) | 86.498 (0.281) | 87.519 (0.286) | *87.937* (0.250) | **88.629** (0.260) |
> | CE     | 1500 | 0.579 (0.021)  | 0.575 (0.020)  | 0.666 (0.015)  | 0.599 (0.015)  | *0.531* (0.018)  | **0.505** (0.016)  |
> |      | 3000 | 0.460 (0.011)  | 0.482 (0.010)  | 0.529 (0.007)  | 0.462 (0.007)  | *0.439* (0.010)  | **0.396** (0.009)  |
>
> In addition, we conducted the additional experiment on the Polarity dataset using pretrained LLM representations. The details of the LLM-based candidate models are provided in our response to Reviewer ho6y (Q1). For your convenience, we also attach the corresponding results below.
>
> | Metric   | Fusion         | Best           | SA-FRL         | SA-cand             | MS          | PEARL                   |
> |----------|----------------|----------------|----------------|----------------|----------------|----------------------------|
> | Acc (%)  | 94.83 (0.75) |  92.95 (0.89) |  95.34 (0.69) |  95.49 (0.70) | *96.41* (0.65) | **96.72** (0.65)           |
> | CE       | 0.139 (0.019)  | 0.209 (0.024)  | 0.165 (0.007)  | 0.130 (0.007)  | *0.100* (0.015)  | **0.091** (0.011)          |
>
>
>
> As shown in the three tables of new conducted experiments, when the candidate models become stronger, the performance of our aggregation method improves accordingly and continues to exhibit numerical superiority over individual candidate models as well as other competing approaches. We hope these additional experiments directly address your concern.

---

### Official Review · Reviewer_m98p · 2025-10-30

**Soundness:** 3
**Presentation:** 3
**Contribution:** 2
**Rating:** 4
**Confidence:** 3

**Summary:**

This paper proposes a new method called Performance-Enhanced Aggregated Representation Learning (PEARL). The main idea is to combine multiple representation learning models to improve performance on downstream tasks (like classification or regression). Instead of picking just one model, PEARL uses a weighted average of predictions from many candidate models. The weights are chosen using cross-validation and a surrogate loss function to make computation easier. The authors provide theoretical guarantees showing that PEARL performs as well as the best possible combination of models and can identify the best models in the candidate set. They test PEARL on several datasets (image, text, and multi-modal) and show it often works better than baseline methods.

**Strengths:**

1. The work appears technically sound. The method is described in detail, and the authors support their claims with theoretical proofs (asymptotic optimality and weight consistency) and extensive experiments on synthetic and real-world datasets.

2. The paper is generally well-written and organized. The two-stage process of PEARL is clearly explained, and the algorithm is presented step-by-step. The examples help illustrate how candidate representations are constructed.

**Weaknesses:**

1. The novelty seems fair. The core idea of combining multiple representation learning models is not well motivated. The explanation in Lines 37-40 does not adequately elaborate on the motivation behind this approach.

2. The experiments, while varied, could be strengthened by including a comparison with more recent or sophisticated ensemble/aggregation methods beyond simple averaging and model selection. This would better situate PEARL's performance within the current landscape.

3. The theoretical results rely on several assumptions. The practical validity or verifiability of these assumptions for real-world data and complex models is not discussed, which is crucial for understanding the real-world applicability of the guarantees.

**Questions:**

The introduction should more clearly elaborate on the necessity of combining multiple representation learning models.

---

> ### Author Response · Authors · 2025-11-21
>
> We sincerely thank you for your constructive comments. We truly appreciate the time and effort you’ve dedicated to reviewing our submission. We hope that our clarifications address your concerns, and we would be happy to provide further discussion or additional details if needed.
>
> > **Weakness 1**:  *The novelty seems fair. The core idea of combining multiple representation learning models is not well motivated. The explanation in Lines 37-40 does not adequately elaborate on the motivation behind this approach.*
>
> **Response 1**: Thank you for your comment. We will strengthen our motivation in the revised version. Our core motivation is that different representation learning methods capture different information, and no single representation learning method is reliably optimal across tasks, domains, or data regimes. Three empirical facts and one toy example below motivate representation aggregation:
>
> - Empirical motivation
>
>   (1) Different representation learning methods tend to specialize in different kinds of information. For example, in vision, Raghu et al. (2021) show that ViTs use early self-attention layers to aggregate more global information, whereas CNNs rely on fixed local receptive fields in lower layers. For language models, different pretrained models and even different layers within the same model capture complementary syntactic and semantic information (Liu et al., 2019).
>
>   (2) Representation performance is unstable across tasks and data distributions. Even the ''best'' representation on one dataset may fail on another—a phenomenon documented by recent large-scale benchmarking studies (Ericsson et al., 2021; Agostinelli et al., 2022).
>
>   (3) Combined representations often improve task performance. Prior works on multi-view, multi-modal, and ensemble feature learning consistently show that combining diverse representations yields more robust predictors. However, these methods lack theoretical guarantees.
>
> - Toy example
>
>   Suppose we have $M = 3$ FRL models that produce $M$ representation sets $z_{(1)}, z_{(2)}, z_{(3)}$, and the ground truth target depends only on the first two: $$  y = g^\star\big(z_{(1)}, z_{(2)}\big) + \varepsilon, $$
>   while $z_{(3)}$ is essentially noise. If we only fit one prediction model per FRL, i.e., $$ \hat g_1(z_{(1)}),\quad \hat g_2(z_{(2)}),\quad \hat g_3(z_{(3)}), $$
>   then $\hat g_1$ and $\hat g_2$ are misspecified (each sees only part of the relevant information, e.g., training a model that uses only $z_{(1)}$ to predict $y$), and $\hat g_3$ is almost useless.
>
>   In our framework, we can derive $J$ candidate representation sets from these $M$ FRLs. For example, if we use the Foundation Set Combinations method in Example 1 of the paper, we construct candidates $$ (1),\ (2),\ (3),\ (1,2),\ (1,3),\ (2,3),\ (1,2,3), $$
>   where “$(1,2)$” denotes the concatenation of $z_{(1)}$ and $z_{(2)}$, etc. The candidate construction creates correctly specified models, such as $(1,2)$ and $(1,2,3)$, out of a starting point where each single-FRL model is misspecified or uninformative. Among these candidates, $(1,2)$ contains exactly the sufficient information for $y$, while the full fusion $(1,2,3)$ adds many redundant or noisy dimensions and is more prone to overfitting. PEARL operates on this candidate family: it learns weights over the $J$ candidates from cross-validated predictions.
>   While in real-world applications, the tasks are more complicated and cannot be illustrated directly by this example, constructing candidates creates better $g$ compared to the original FRL models. As illustrated by our experimental results, our aggregation method can outperform the simple fusion model and is more robust than selecting a single candidate from the $J$ candidates.
>
>
> References
>
> [1] Andrea Agostinelli, Michal P´andy, Jasper Uijlings, Thomas Mensink, and Vittorio Ferrari. (2022). "How stable are transferability metrics evaluations?" In European Conference on Computer Vision, 303–321.
>
> [2] Linus Ericsson, Henry Gouk, and Timothy M. Hospedales. (2021). "How well do self-supervised models transfer?" In Proceedings of the IEEE/CVF Conference on Computer Vision and Pattern Recognition, 5414–5423.
>
> [3] Liu, N. F., Gardner, M., Belinkov, Y., Peters, M. E., & Smith, N. A. (2019). "Linguistic Knowledge and Transferability of Contextual Representations." Proceedings of NAACL-HLT 2019.
>
> [4] Raghu, M., Unterthiner, T., Kornblith, S., Zhang, C., & Dosovitskiy, A. (2021). "Do Vision Transformers See Like Convolutional Neural Networks?" NeurIPS.

---

> > ### Author Response · Authors · 2025-11-21
> >
> > > **Weakness 2**: *The experiments, while varied, could be strengthened by including a comparison with more recent or sophisticated ensemble/aggregation methods beyond simple averaging and model selection. This would better situate PEARL's performance within the current landscape.*
> >
> > **Response 2**：Thank you for this valuable suggestion.  Following your suggestion, we conducted additional experiments incorporating a nonlinear neural network (NN) aggregator, which is a sophisticated ensemble/aggregation approaches commonly used in deep learning.
> >
> > Let $\hat g_j(x)$ denote the prediction vector of the $j$-th candidate model on input $x$. For each training sample $x_i$, we form the concatenated vector $v_i = \big(\hat g_1(x_i),\dots,\hat g_J(x_i)\big)$ and train an MLP $h_\theta$ to predict $y_i$ from $v_i$. We evaluated this nonlinear baseline, as denoted by NN-aggregator, on CIFAR-10 and the Polarity dataset. The results are shown below:
> >
> > | Dataset | Metric | n    | PEARL                 | NN-aggregator         |
> > |---------|--------|------|-----------------------|----------------------|
> > | CIFAR-10 | Acc(%) | 300  | 53.83 (0.83)        | 44.81 (1.24)        |
> > |         |        | 600  | 57.03 (0.52)        | 50.83 (0.97)        |
> > |         |        | 1500 | 60.19 (0.38)        | 56.56 (0.62)        |
> > |         |        | 3000 | 61.93 (0.36)        | 60.25 (0.53)        |
> > |         | CE | 300  | 1.282 (0.023)        | 3.331 (0.174)       |
> > |         |        | 600  | 1.190 (0.012)        | 2.754 (0.178)       |
> > |         |        | 1500 | 1.104 (0.007)        | 1.674 (0.070)       |
> > |         |        | 3000 | 1.054 (0.006)        | 1.293 (0.023)       |
> > | Polarity Dataset | Acc(%) | 1400    | 88.873 (1.173)       | 86.029 (1.264)      |
> > |         | CE | 1400    | 0.291 (0.013)        | 0.358 (0.039)
> >
> > Across both datasets and all sample sizes, the nonlinear neural aggregator consistently underperforms PEARL. In addition, our linear weighting provides better interpretability of candidate representations, whereas the NN-based ensemble operates as a black-box.
> >
> > > **Weakness 3**: *The theoretical results rely on several assumptions. The practical validity or verifiability of these assumptions for real-world data and complex models is not discussed, which is crucial for understanding the real-world applicability of the guarantees.*
> >
> > **Response 3**: Thank you for your comment on assumptions. We summarize them below in terms of the practical interpretations of assumptions.
> >
> > (1) Assumptions linking surrogate loss and the original loss (C.1). The assumption is standard in surrogate-risk theory (Dai et al., 2022) and holds for commonly used losses (Shen et al., 2003; Shen & Wong, 2010). If the assumption was violated, the surrogate-based training procedure would be unreliable, not just for PEARL.
> >
> > (2) Assumptions on the surrogate loss and its variability (C.2, C.3, C.4, C.7). These assumptions require that the surrogate loss has bounded variance and mild smoothness properties. We illustrate that widely used surrogate losses satisfy C.4 and C.7.  Assumption C.2 and its similar assumptions are routinely used in analyzing cross-validation, sample splitting, and generalization of deep models. Modern neural networks with standard regularization (batch-norm, weight decay, early stopping) naturally avoid arbitrarily large loss fluctuations, and therefore C.3 holds.
> >
> > (3) Assumptions on convergence rates and CV predictors (C.5, C.6, C.8). C.5 requires that both the limiting predictor and its CV estimators converge at comparable rates. This is a standard requirement in cross-validation theory and holds for deep models whenever training is stable and predictors are trained on sufficiently large samples. Importantly, we do not require a specific rate (e.g., $n^{-1/2}$) since we do not restrict architectures or optimization methods. Assumptions involving the population risk (e.g., C.6 and C.8) are standard in model averaging theory, and the verifiability of similar assumptions has already been thoroughly analyzed in prior work (Wan et al., 2010).
> >
> > Reference
> >
> > [1] Wan, Alan TK, Xinyu Zhang, and Guohua Zou. (2010). "Least squares model averaging by Mallows criterion." Journal of Econometrics, 156(2): 277-283.
> >
> > >**Question**: *The introduction should more clearly elaborate on the necessity of combining multiple representation learning models.*
> >
> > **Response**:
> > Thank you for pointing this out. In the revision, we will substantially expand the motivation in the Introduction to explain more clearly why combining multiple representation learning models is necessary. We will incorporate both the empirical evidence and the toy example discussed in our response to Weakness 1, so that the necessity and advantages of aggregating multiple representations will be clearer.

---

### Official Review · Reviewer_ho6y · 2025-11-03

**Soundness:** 2
**Presentation:** 3
**Contribution:** 3
**Rating:** 4
**Confidence:** 2

**Summary:**

This paper introduces PEARL, a general framework that combines multiple representation learning methods to improve downstream task performance. The authors propose a two-stage model averaging approach—first training diverse foundational representation learning (FRL) models, then aggregating their outputs using cross-validated, surrogate-loss–based weighting for computational efficiency and theoretical guarantees of asymptotic optimality and weight consistency. Experiments verify the proposal's performance.

**Strengths:**

1. The paper is clearly structured and well written, with a logical flow from motivation to theoretical results and experiments.
2. The paper presents a novel and well-motivated framework, PEARL, that integrates multiple representation learning methods under a unified model-averaging paradigm.

**Weaknesses:**

1. Training cost: The computational cost deserves further discussion. Since the selection process relies on K-fold cross-validation, the overall training expense may exceed that of conventional single-model approaches.
2. Efficiency evidence: Although the authors claim that PEARL achieves computational efficiency through the use of surrogate losses and linear weighting, the paper provides limited empirical or complexity analysis to substantiate this claim. Including runtime comparisons or scalability analyses would strengthen the argument.
3. Novelty and related work: While the paper presents a theoretically grounded and flexible aggregation framework, the central idea of combining multiple representations via weighted model averaging is conceptually related to established ensemble and model averaging methods [1, 2]. The authors could enhance the originality claim by clarifying how PEARL differs from prior unified model-averaging frameworks—particularly whether its surrogate-loss formulation or asymptotic guarantees contribute fundamentally new theoretical insights.

[1] Least Squares Model Averaging (LSMA)
[2] Model Averaging Prediction by K-fold Cross-Validation

**Questions:**

I wonder whether this method could be extended to large language model (LLM) settings rather than being limited to small-scale scenarios. If the framework can guide the selection of suitable training recipes or adaptation strategies for LLM agents across different tasks, its impact would be even more substantial.

Note: I am not deeply familiar with this specific research area, so my evaluation may carry some bias. I will adjust my final score based on insights from other reviewers.

---

> ### Author Response · Authors · 2025-11-21
>
> We sincerely thank you for your constructive comments. We truly appreciate the time and effort you’ve dedicated to reviewing our submission. We hope that our clarifications address your concerns, and we would be happy to provide further discussion or additional details if needed.
> > **Weakness 1**: *Training cost: The computational cost deserves further discussion. Since the selection process relies on K-fold cross-validation, the overall training expense may exceed that of conventional single-model approaches*.
>
> **Response 1**：Thank you for your comment. We agree that CV introduces additional overhead. However, for PEARL, the extra cost is practically manageable for the following reasons:
>
> - Our CV process does not retrain the representation models, but only retrains the downstream task models. Training the foundation representations is by far the most expensive step in PEARL and in all competing baselines. For downstream task learning, for each candidate $j$ and each fold $k$, PEARL trains the predictors $\widetilde{g}_j^{(-k)}$ (e.g., a simple MLP or linear head). These models are fast to train, which keeps the overall retraining time for the predictors low.
> - CV can be efficiently parallelized or approximated. The $K$ downstream trainings can be parallelized across CPU cores, and thus the entire CV step will be largely accelerated. In addition, PEARL can adopt approximate CV techniques, which further reduce CV cost by reusing computations across folds (Xu et al., 2012; Wilson et al., 2020).
> - Weight optimization is convex and computationally fast, since the weight criterion is based on the surrogate loss instead of the original loss.
>
> To address your concern empirically, we report the runtime of PEARL on CIFAR-10 and the Polarity dataset using two types of downstream models $g$:  logistic regression and a small MLP. For each setting, we report the runtime of (i) CV ($K=10$, 10 CPU cores), (ii) downstream task training time ($\widehat{g}_j$), and (iii) weight-optimization time. As shown below, the overall runtimes including CV runtime remain small.
>
> |   Dataset |   M |   J | Type of g   | CV (parallel) [s] |  Downstream Training [s] | Weight Optimization [s] | Total Runtime[s] |
> | ---------:| ---:| ---:| ----------- | -----------------:| --------------------:| ----------------------:| ---------:|
> |CIFAR-10 |  4 |  15 | Logistic      |               4.953  |         5.195 |  0.129 |    10.277    |
> |           |     |     | MLP         |             7.680 |         12.891 |   0.152 |       20.723 |
> |  Polarity |  6 |  6 | Logistic      |               7.421 |                 2.541  |                    0.004 |       9.966 |
> |           |     |     | MLP         |               8.586|                 4.471 |                   1.169 |       14.226 |
>
>
> References:
>
> [1] Xu, Ganggang, and Jianhua Z. Huang. (2012). "Asymptotic optimality and efficient computation of the leave-subject-out cross-validation." The Annals of Statistics, 40(6): 3003-3030.
>
> [2] Wilson, Ashia, Maximilian Kasy, and Lester Mackey. (2020). "Approximate cross-validation: Guarantees for model assessment and selection." International Conference on Artificial Intelligence and Statistics, 4530-4540.
>
> > **Weakness 2**：*Efficiency evidence: Although the authors claim that PEARL achieves computational efficiency through the use of surrogate losses and linear weighting, the paper provides limited empirical or complexity analysis to substantiate this claim. Including runtime comparisons or scalability analyses would strengthen the argument.*
>
> **Response 2**：Thank you for your comments. Following your suggestion, we present complexity analysis and  empirical runtime comparisons below.
>
> - Complexity analysis. Surrogate losses enable polynomial-time optimization and avoid NP-hard objectives. For example, optimizing the 0–1 loss for two-class classification is NP-hard. By cross-entropy as a surrogate loss in our method, the weight-optimization process in PEARL has computational complexity $O(nJ)$ per iteration with $J$ candidates.
>
>   For nonlinear weighting, we take neural network (NN) aggregator as example.  Specifically, let $\hat g_j(x)$ denote the prediction vector of the $j$-th candidate model on input $x$. For each training sample $x_i$, we form the concatenated vector $v_i = \big(\hat g_1(x_i),\dots,\hat g_J(x_i)\big)$ and train an two layer MLP $h_\theta$ to predict $y_i$ from $v_i$. The weight optimization for NN aggregator with hidden size $H$ requires $O(nJH)$ per iteration. In addition, linear weighting yields better interpretability.
>
> - Runtime comparisons. We compare the runtime of one-time weight optimization of PEARL with linear weighting and NN aggregator on CIFAR-10 dataset. From the table below, we can see that linear weighting optimization is much faster than NN aggregator optimization.
>
> |  | Linear weighting (s) | NN aggregator (s)|
> | -------- | -------- | -------- |
> | CIFAR-10     | 0.129     | 2.235     |

---

> > ### Author Response · Authors · 2025-11-21
> >
> > > **Weakness 3**: *Novelty and related work: While the paper presents a theoretically grounded and flexible aggregation framework, the central idea of combining multiple representations via weighted model averaging is conceptually related to established ensemble and model averaging methods [1, 2]. The authors could enhance the originality claim by clarifying how PEARL differs from prior unified model-averaging frameworks—particularly whether its surrogate-loss formulation or asymptotic guarantees contribute fundamentally new theoretical insights.
> > > [1] Least Squares Model Averaging (LSMA) [2] Model Averaging Prediction by K-fold Cross-Validation*
> >
> > **Response 3**: Thank you for this important point. We agree that LSMA [1] and K-fold model averaging [2] are two foundational, theoretically guaranteed approaches in the model-averaging literature. However, prior theoretically grounded model-averaging methods primarily rely on **specific parametric assumptions** (e.g., linear or quasi-likelihood models) and **convex or smooth losses** (typically quadratic). In contrast, PEARL is able to handle **heterogeneous deep learning representations and non-convex losses** via estimating weights on the surrogate loss, which is the primary theoretical distinction from existing model-averaging works.
> >
> > In the machine learning community, K-fold CV–based aggregation (similar to stacking) is empirically model-agnostic, but **the theoretical guarantees for its efficiency or optimality remain largely unexplored**. Our paper addresses this gap. As a result, PEARL provides both empirical effectiveness and theoretical guarantees, enabling practitioners to apply such aggregation methods to advanced deep learning methods.
> >
> > Moreover, our framework is highly general and can include classical statistical model averaging methods as special cases within our candidate models.
> >
> >
> >
> > > **Q1**: *I wonder whether this method could be extended to large language model (LLM) settings rather than being limited to small-scale scenarios. If the framework can guide the selection of suitable training recipes or adaptation strategies for LLM agents across different tasks, its impact would be even more substantial.*
> >
> > **Response to Q1**: Thank you for this valuable question. We agree that extending PEARL to LLMs is both natural and impactful. We therefore conducted an additional experiment applying PEARL to a polarity classification dataset with LLM-based representations.
> >
> > Specifically, we consider three pretrained LLMs as FRL models:  ```"Qwen/Qwen2.5-1.5B-Instruct"```, ```"meta-llama/Llama-3.2-1B-Instruct"```, and ```"openai-community/gpt2-xl"```. The first two are commonly used open-source instruction-tuned LLMs, and the third is a widely used GPT-series model available via the openai-community repository. In this setting, we construct candidate models based on the representations induced by these LLMs, resulting in $J=2^3-1=7$ candidates.  The results are reported in the following table. They show that PEARL can directly improve LLM-based classification performance without any modification to the LLMs themselves.
> >
> > | Metric   | Fusion         | Best           | SA-FRL         | SA-cand             | MS          | PEARL                   |
> > |----------|----------------|----------------|----------------|----------------|----------------|----------------------------|
> > | Acc (%)  | 94.83 (0.75) |  92.95 (0.89) |  95.34 (0.69) |  95.49 (0.70) | *96.41* (0.65) | **96.72** (0.65)           |
> > | CE       | 0.139 (0.019)  | 0.209 (0.024)  | 0.165 (0.007)  | 0.130 (0.007)  | *0.100* (0.015)  | **0.091** (0.011)          |
> >
> > Note: MS is model selection using cross-validation based on our candidates, which needs our candidate construction procedure and the CV process.
> >
> > In summary, our current framework allows us to combine multiple pretrained LLM representations for a target task with theoretical guarantees. Extending this idea, different training recipes or adaptation strategies for LLM agents could be instantiated as distinct candidate models, although doing so would require additional design beyond the scope of our current framework. In such an extended setting, the learned weights may provide insights into which strategies are better aligned with the target task.

---

### Author Response · Authors · 2025-12-02
**Summary of revisions**

Dear Area Chair,

We sincerely thank you and the four reviewers for the careful handling and thorough reading of our submission ID 23194, “Performance-Enhanced Aggregated Representation Learning (PEARL)”. We have provided point-by-point responses to all reviewer comments and have revised the manuscript accordingly, as well as substantially expanded the supplementary material. In particular, we have added a new set of experiments, including those based on large language model (LLM) representations. Below is a short structured overview of what we did:

1. Motivation and novelty (Introduction revised).

(1) We revised the introduction to more clearly explain why combining multiple representations is necessary, emphasizing that different representation learners capture complementary information and no single representation is uniformly optimal across tasks, data regimes, or modalities.

(2) We added additional empirical evidence and a toy example to the responses, and clarified in the paper how PEARL differs from existing model-averaging methods: unlike classical approaches with theoretical guarantees (e.g., linear/quasi-likelihood with convex loss), PEARL can handle heterogeneous deep models and generalized loss functions while still providing asymptotic guarantees.

2. Additional numerical studies (Section 4 and Appendices B and E).

In response to comments on training cost, nonlinear aggregation methods, stronger candidate models, and LLM-based representations, we added a large set of new experiments to the rebuttal, Section 4 and Appendices B and E in our paper. We show that:

(1) The computational cost of PEARL is moderate relative to standard training pipelines.

(2) PEARL outperforms nonlinear neural network aggregators (NN-aggregators) while remaining more interpretable.

(3) When we strengthen the candidate models (e.g., using pretrained ResNets/ViTs), PEARL still improves over strong individual candidates and simple fusion schemes.

(4) PEARL can be directly applied to pretrained LLM-based representations, where it consistently improves performance without modifying the underlying LLMs.

3. Theoretical assumptions and asymptotic guarantees.

(1) We expanded the discussion of our assumptions, explaining that widely used surrogate loss functions in practice satisfy the key conditions needed for our theory.

(2) We further emphasized the significance of our asymptotic optimality result: PEARL is asymptotically superior to simple fusion, equal-weight combinations, and any single candidate model. Since selecting a single candidate corresponds to using a one-hot weight vector, our optimality over the full weight space automatically implies superiority over all model-selection-type choices as a special case.

We are very grateful for the opportunity to have our work considered for the ICLR program, and we deeply appreciate your time and effort in evaluating both the original submission and our detailed responses.

Best regards,

Authors

---

### Meta-Review · Area_Chair_63UL · 2026-01-01

**Summary:**

The most critical issue of the submission is the lack of sufficient novelty, as noted by all reviewers, and I agree with them. I recommend rejecting the paper.

**Reviewer Concerns:**

* The lack of sufficient novelty is noted by all reviewers, and I also agree with them.

* Only Reviewer u8HM gave a positive score but the reviewer was not excited about the paper either, and did not provide any strong positive evidence to accept this paper. The reviewer had a problem with the novelty of the work and I agree with the reviewer.

* The additional experiment responding to Reviewer m98p (comparing with only one recent mehod) is not satisfactory.

* The overall performance enhancement exhibited by the paper is not significant enough.

**Reviewer Scores:**

* Reviewer ho6y: I do not think the reviewer would have increased the score.

* Reviewer m98p: I do not think the reviewer would have increased the score.

* Reviewer ioJk: I do not think the reviewer would have increased the score.

* Reviewer u8HM: I do not think the reviewer would have increased the score any further.

---

### Decision · Program_Chairs · 2026-01-26

Reject